# Regulatory network structure determines patterns of intermolecular epistasis

**Mato Lagator[1], Srdjan Sarikas[1], Hande Acar[1], Jonathan P Bollback[1,2]\*, Călin C Guet[1]\***

[1]Institute of Science and Technology Austria, Klosterneuburg, Austria; [2]Institute of Integrative Biology, University of Liverpool, Merseyside, United Kingdom

**Abstract** Most phenotypes are determined by molecular systems composed of specifically interacting molecules. However, unlike for individual components, little is known about the distributions of mutational effects of molecular systems as a whole. We ask how the distribution of mutational effects of a transcriptional regulatory system differs from the distributions of its components, by first independently, and then simultaneously, mutating a transcription factor and the associated promoter it represses. We find that the system distribution exhibits increased phenotypic variation compared to individual component distributions - an effect arising from intermolecular epistasis between the transcription factor and its DNA-binding site. In large part, this epistasis can be qualitatively attributed to the structure of the transcriptional regulatory system and could therefore be a common feature in prokaryotes. Counter-intuitively, intermolecular epistasis can alleviate the constraints of individual components, thereby increasing phenotypic variation that selection could act on and facilitating adaptive evolution.

DOI: https://doi.org/10.7554/eLife.28921.001

## Introduction

Distributions of mutational effects (DMEs) and the nature of the interactions among mutations (epistasis) critically determine evolutionary paths and outcomes (*Eyre-Walker and Keightley, 2007*; *de Visser and Krug, 2014*). DMEs are central to a range of fundamental questions in evolutionary biology (*Halligan and Keightley, 2009*), including understanding the origins of novel traits (*Soskine and Tawfik, 2010*), evolution of sex and recombination (*Otto and Lenormand, 2002*), and maintenance of genetic variation (*Charlesworth et al., 1995*). In contrast to selective constraints, which act on the variation already present in a population, biophysical laws and molecular mechanisms that define how a molecular system functions constrain the access to phenotypic variation through mutation (*Camps et al., 2007*; *Wagner, 2011*), and in doing so determine the shape of the DME (*Fontana and Buss, 1994*).

Even though most phenotypes are determined by underlying molecular systems that consist of multiple specifically interacting molecular components, direct and systematic experimental estimates of DMEs have been limited to the two extremes only: either at the level of the whole organism, obtained in mutation accumulation studies (*Halligan and Keightley, 2009*); or at the level of individual components, such as proteins (*Wang et al., 2002*; *Bershtein et al., 2006*; *Sarkisyan et al., 2016*) and DNA-binding sites for transcription factors (*Kinney et al., 2010*; *Shultzaberger et al., 2012*; *Yun et al., 2012*; *Metzger et al., 2015*), determined through direct mutagenesis. Knowing only the effects of mutations in individual molecular components might be insufficient to understand how the whole system evolves, as recent studies focusing on the interaction of mutations in two components of a molecular system uncovered the existence of pervasive intermolecular epistasis (*Anderson et al., 2015*; *Podgornaia and Laub, 2015*). Because of the large mutational space of proteins, these studies focused only on mutations in specific residues that lie at the interface between

**\*For correspondence:**
J.P.Bollback@liverpool.ac.uk (JPB);
calin@ist.ac.at (CCG)

**Competing interests:** The authors declare that no competing interests exist.

the two interacting molecules. In contrast, addressing the more general question of how intermolecular epistasis shapes DMEs of molecular systems is only possible by experimentally tackling the nearly prohibitive space of possible mutational combinations, which even for single components is conceivable only in rare cases (*Sarkisyan et al., 2016*). Here, by using one of the best understood transcriptional regulation systems, we experimentally ask how the DME of a system differs from the DMEs of its constitutive components.

To address this question, we used a simple gene regulatory system based on the canonical Lambda bacteriophage switch (*Ptashne, 2011*), consisting of three components – the $\sigma^{70}$RNA polymerase complex (together, we refer to them as RNAP), transcription factor CI (*trans*-element), and the $P_R$ cis-element that contains the overlapping DNA-binding sites of the two proteins (*Figure 1*). These three molecular components interact to produce a quantitative phenotype: gene expression. Specifically, we used a genetic system in which a strong promoter $P_R$ controls the expression of a yellow fluorescence protein (*yfp*) and is repressed by the CI repressor, which we placed under the inducible promoter $P_{TET}$ (*Figure 1B,C*). This system exhibits high *yfp* expression in the absence of CI, where the level of expression is determined solely by RNAP binding. However, in the presence of CI, achieved by the induction of the $P_{TET}$ promoter, the system is strongly repressed (*Figure 2A*). We find that, even in such a simple transcriptional regulatory system, the DME of the system differs

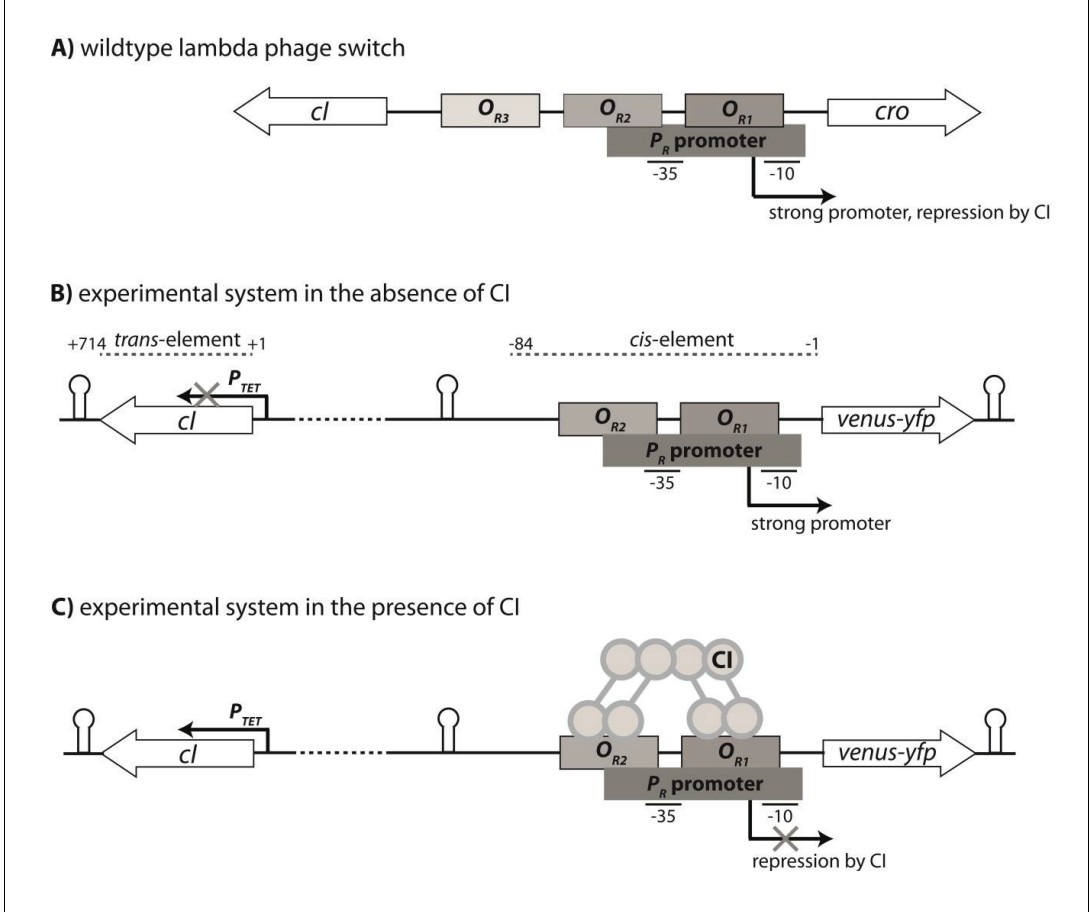

**Figure 1.** Genetic organization of the Lambda phage switch and the experimental system. (**A**) The Lambda phage switch consists of two transcription factors - CI and Cro; two promoters - a strong promoter $P_R$ and a weak promoter $P_{RM}$ (not shown); and three operator sites - $O_{R1}$, $O_{R2}$, and $O_{R3}$. (**B**) The experimental synthetic system, where *cro* was substituted with the fluorescence marker gene (*venus-yfp*) under control of $P_R$. The $P_{RM}$ promoter was removed by the removal of parts of $O_{R3}$. Located 500 bp away on the reverse strand and separated by a terminator sequence is the *cl* gene under control of an inducible $P_{TET}$ promoter. CI, the *trans*-element, is 714 bp; the *cis*-element is 84 bp long. (**C**) CI dimers bind cooperatively to $O_{R1}$ and $O_{R2}$, leading to repression of the $P_R$ promoter.
DOI: https://doi.org/10.7554/eLife.28921.002

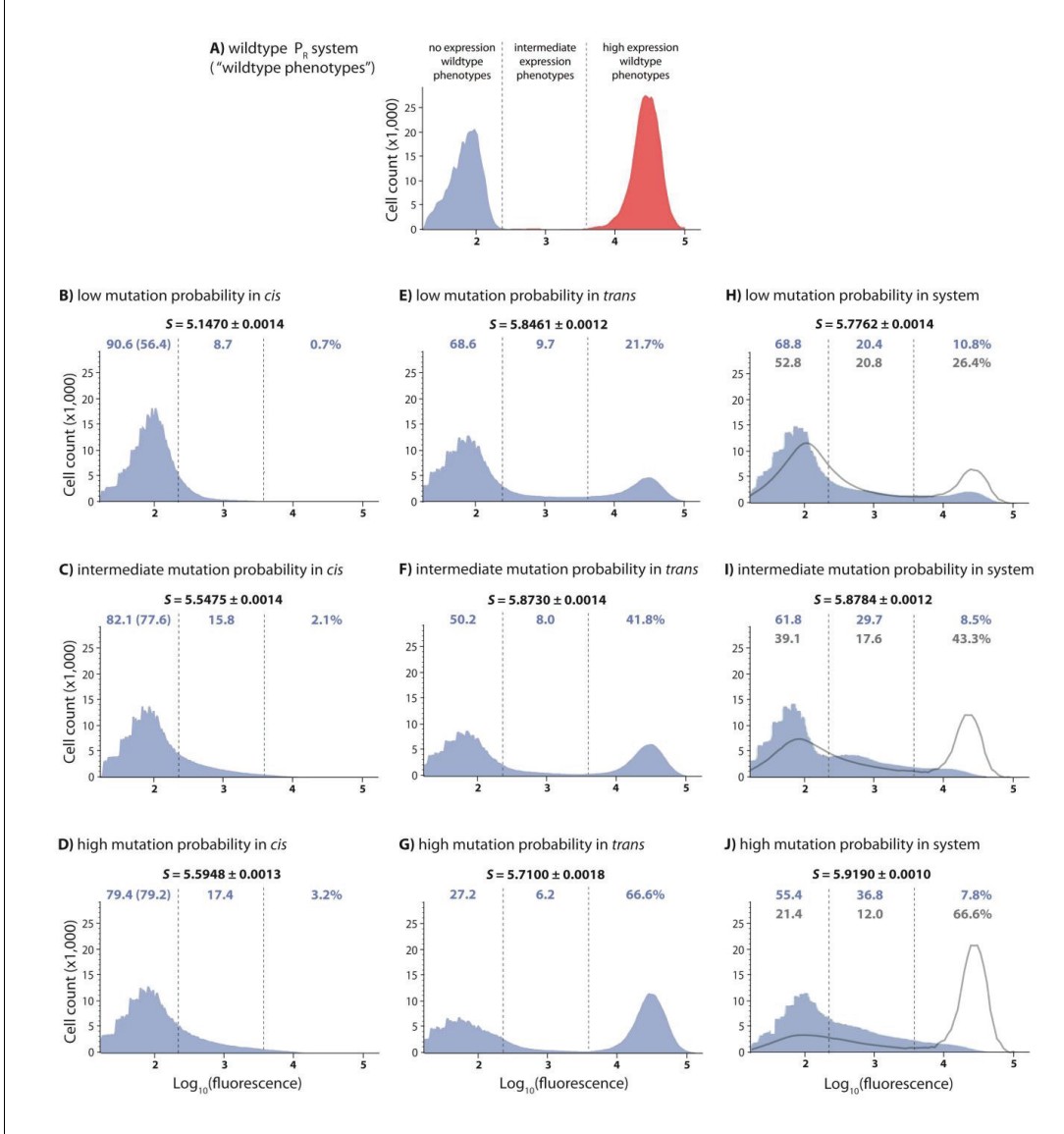

**Figure 2.** DMEs of the whole system are more evenly distributed than the individual component DMEs. In the experimental system, CI acts as a tight repressor. The distributions of fluorescence are shown in the absence of CI (red) and in the presence of CI (blue). Each distribution was obtained by measuring fluorescence of two independent measurements of 500,000 cells by flow cytometry, which were then pooled together. The dashed lines separate three categories of phenotypes – 'no expression' phenotypes (corresponding to repressed wildtype); 'high expression' phenotypes (corresponding to the wildtype in the absence of CI); and 'intermediate' phenotypes. No expression and high expression categories are defined to include >99.9% of the wildtype fluorescence distribution in the presence and in the absence of CI, respectively. The Shannon entropy (S) is used to estimate how uniform each distribution is across the entire range of possible expression levels. The associated standard deviation (±) is given for each S value. Blue numbers are percentage of counts in each category in the presence of CI. Numbers in parentheses are percentage of counts excluding the estimated percentages of uniquely transformed individuals carrying the wildtype genotype (see Materials and methods). The naïve additive convolution prediction for each system library and the associated predictions for the frequency of mutants in each category are shown in grey. Pearson's Chi-squared test was used to assess the difference between the observed and the convolution-predicted frequency of mutants in each category (low: $\chi^2_{(2)}=8.20$; p<0.05; intermediate: $\chi^2_{(2)}=32.26$; p<0.0001; and high mutation frequency library: $\chi^2_{(2)}=74.51$; p<0.0001). The distributions of the effects of mutations for the *cis*-element, the *trans*-element, and the whole system in the absence of CI are shown in *Figure 2—figure supplement 1*. *Figure 2—figure supplement 2* shows distributions of the effects of 150 single point mutations in the *cis*- and the *trans*-elements. Statistical significance of the differences in entropy values between the mutant libraries is shown in *Figure 2—source data 3*. Flow cytometry measurements of 20 individual isolates from each library are shown in *Figure 2—figure supplements 3*, *4* and *5*, the analysis of which was used to demonstrate that gene expression noise is constant (*Figure 2—source data 1*). Convolutions for each mutation probability performed with the knowledge of the genetic regulatory structure of the system are shown in *Figure 2—figure supplement 6*, while *Figure 2—figure supplement 7* provides an explanation of how convolutions were performed. The outcome of the test for how sensitive the shapes of distributions are to the number of sampled individuals is shown in *Figure 2—*

*Figure 2 continued*

**source data 4**, while the confirmation that the mutagenesis protocol resulted in expected distributions of the number of mutations are shown in **Figure 2—source data 2**.

DOI: https://doi.org/10.7554/eLife.28921.003

The following source data and figure supplements are available for figure 2:

**Source data 1.** Gene expression noise is constant.

DOI: https://doi.org/10.7554/eLife.28921.011

**Source data 2.** Sequencing 40 isolates from each *cis*- and *trans*-element library confirms the predicted distribution of the number of mutations.

DOI: https://doi.org/10.7554/eLife.28921.012

**Source data 3.** Differences between calculated entropy estimates are statistically significant.

DOI: https://doi.org/10.7554/eLife.28921.013

**Source data 4.** Observed distributions accurately describe phenotypic distributions of possible mutations.

DOI: https://doi.org/10.7554/eLife.28921.014

**Figure supplement 1.** DMEs for *cis*-element, *trans*-element, and system libraries in the absence of CI.

DOI: https://doi.org/10.7554/eLife.28921.004

**Figure supplement 2.** Distribution of single mutation effects in 150 random system double mutants and their corresponding single mutants.

DOI: https://doi.org/10.7554/eLife.28921.005

**Figure supplement 3.** Mutant isolates from the low mutation probability libraries.

DOI: https://doi.org/10.7554/eLife.28921.006

**Figure supplement 4.** Mutant isolates from the intermediate mutation probability libraries.

DOI: https://doi.org/10.7554/eLife.28921.007

**Figure supplement 5.** Mutant isolates from the high mutation probability libraries.

DOI: https://doi.org/10.7554/eLife.28921.008

**Figure supplement 6.** Mathematical predictions that account for the genetic regulatory structure accurately describe the system DME.

DOI: https://doi.org/10.7554/eLife.28921.009

**Figure supplement 7.** Predicting the system DME based on convolving component DMEs.

DOI: https://doi.org/10.7554/eLife.28921.010

unexpectedly from the individual component DMEs, and that most of this difference can be directly attributed to the genetic regulatory structure of the system.

# Results

To determine the DMEs of individual components in our system, we performed direct mutagenesis on the *cis*- and the *trans*-element independently. For each component, we created libraries with approximately 0.01, 0.04, and 0.07 mutations per nucleotide (low, intermediate, and high mutation probability libraries, respectively). Due to the different sizes of the two components (84 bp for *cis* and 714 bp for *trans*), mutants in cis-element libraries contained on average 1, 3, and 6 mutations, whereas mutants in trans-element libraries contained on average 7, 28, and 49 mutations, respectively. We did not mutate the $\sigma^{70}$ – RNAP complex due to its cell-wide pleiotropic effects. For assessing the DMEs of the system, we created three additional 'system mutant libraries' by combining *cis*- and *trans*-element mutant libraries of the same mutation probability. Each library consisted of more than 30,000 uniquely transformed individuals, and we estimated the corresponding DME for each library by measuring fluorescence, that is gene expression, of 1 million randomly sampled individuals by flow cytometry. We quantified the differences in DMEs in two ways. First, by observing the frequency of mutants in three biologically meaningful categories (**Figure 2A**): (i) mutants that are indistinguishable from the wildtype without the CI repressor ('high expression phenotypes'); (ii) mutants indistinguishable from the wildtype with CI ('no expression phenotypes'); and, importantly for our argument, (iii) mutants with expression levels that the wildtype cannot achieve ('intermediate expression phenotypes'). Second, by calculating the Shannon entropy, which quantifies how uniformly the mutants cover the entire range of possible gene expression phenotypes.

## DMEs of the *cis*- and the *trans*-element are categorically different

The difference between the effects of mutations in the *cis*- (**Figure 2B,C,D**) and the *trans*-element (**Figure 2E,F,G**) is unambiguous. Most *cis*-element mutants, in the presence of CI, have low

expression (*Figure 2B,C,D*). Such low expression can result either from sufficient CI repressor binding and/or impaired RNAP binding. The frequency of mutants with an intermediate expression phenotype in the presence of CI increases with the average number of mutations, a pattern that is also observed in cis-element libraries in the absence of CI (*Figure 2—figure supplement 1B,C,D*). In contrast, the effects of mutations in the *trans*-element have a distinctive bimodal distribution (*Figure 2E,F,G*). The two peaks in the distribution correspond to the expression levels of the wild-type population in the absence and presence of CI, revealing that the majority of CI mutants are either fully or close to fully functional, or completely inactive on the wildtype *cis* background. Furthermore, increasing the average number of mutations in the *trans*-element decreases the frequency of intermediate expression phenotypes. As the shape of the DME is determined by the underlying biophysical and mechanistic constraints that limit the phenotypic variation accessible through mutation, we conclude that the *cis*- and the *trans*-element have categorically different constraints. This categorical difference is best observed in a direct comparison between high mutation *cis* (*Figure 2D*) and low mutation *trans* (*Figure 2E*) libraries, which have approximately the same average number of mutations.

In order to further demonstrate that this categorical difference is not an artifact of the different number of mutations introduced into each element, we show that the same general trend is evident when comparing the effects of 150 random single point mutations of known identity in each, the *cis*- and the *trans*-element (*Figure 2—figure supplement 2A,B*). These measurements were done at the population level, in a plate reader. Point mutations in the *cis*-element, when only their effect on RNAP binding is measured, show a high frequency of intermediate expression levels. This is in agreement with other studies of prokaryotic (*Kinney et al., 2010*) and eukaryotic (*Shultzaberger et al., 2012*) DNA-binding sites. Similarly, we find a bimodal distribution of single mutation effects in the *trans*-element CI, which has previously been reported for other transcription factors (*Pakula et al., 1986*; *Markiewicz et al., 1994*) and enzymes (*Jacquier et al., 2013*), and may be a common feature of proteins that are close to their optimum (*Soskine and Tawfik, 2010*; *Bataillon and Bailey, 2014*).

## Mutating the whole system increases phenotypic variation

The DMEs for the system, in which both the *cis*- and the *trans*-element are mutated simultaneously, show a surprising pattern: a higher frequency of intermediate phenotypes compared to either of the individual component DMEs (*Figure 2H,I,J*). This pattern can also be observed in the library of 150 random system double mutants (*Figure 2—figure supplement 2*), which consist of a unique combination of the previously described point mutations in cis and in trans (comparing system to *cis*: $D_{KS}$ = 0.39, p<0.0001; system to *trans*: $D_{KS}$ = 0.66, p<0.0001). Furthermore, the frequency of intermediate phenotypes increased with the average number of mutations (*Figure 2*). At intermediate and high mutation probabilities, the Shannon entropy of the system DMEs was also greater than the entropy of either of its constitutive components (*Figure 2*; *Figure 2—source data 3*), indicating that mutating the whole system gives rise to a more uniform range of possible phenotypes. The existence of a difference between system DMEs in the absence (*Figure 2—figure supplement 1H,I,J*) and in the presence of repressor CI (*Figure 2H,I,J*) indicates that a substantial portion of mutant CIs exhibit binding to the mutated *cis*-element backgrounds, and are thus functional repressors that specifically recognize operator mutants.

We tested if the observed differences between the system and the component DMEs might arise from differences in the gene expression noise of mutants - if the system mutants have greater noise, they could also lead to an increase in the frequency of intermediate phenotypes. However, this is unlikely as, for every mutation probability, we observed no difference in gene expression noise in our flow cytometry measurements (measured as the coefficient of variation) between 20 randomly selected system, *cis*, and *trans* mutants in the presence of CI (*Figure 2—figure supplement 3*, *4* and *5*). Furthermore, gene expression noise was constant between all 180 random isolates (60 isolates from each of the three mutation probabilities) irrespective of their mutation probability (*Figure 2—source data 1*).

## Intermolecular epistasis drives the increase in phenotypic variation

We wanted to understand if the observed increase in the abundance of intermediate phenotypes when the whole system mutates (*Figure 2*) can be attributed to epistatic interactions between mutations in the *cis*- and the *trans*-element. Since we use $\log_{10}$ of expression as our phenotype of interest, we calculate epistasis between mutations in the two components of the system (what we call intermolecular epistasis) as the deviation from the additive prediction based on single component effects. The additive null prediction for interactions between mutations, when considering only three possible categories of mutational effects ('no', 'intermediate', and 'high expression'), is shown in *Table 1*. The standard approach of extending these predictions to whole distributions requires a convolution of individual component DMEs (*Orr, 2003*; *Lee et al., 2010*). To obtain this 'naïve' null prediction, we convolved the observed *trans*-element DME (shown in *Figure 2E,F,G*) with the distribution showing how mutations in cis alter wildtype expression (for further details, see Materials and methods). Note that effectively no mutations in cis or *trans* decrease expression relative to the wildtype, resulting in a 'naïve' system prediction exhibiting an increase in the frequency of mutants with high expression phenotypes, as seen in *Figure 2H,I,J*. We find that 'naïve' convolution predictions, which are carried out in the absence of any knowledge of the genetic regulatory structure of the system, are significantly different from the observed system distributions (*Figure 2H,I,J*), suggesting the existence of intermolecular epistasis between the two components.

To further show that intermolecular epistasis between the *cis*- and the *trans*-element is a common feature of our system that shapes DMEs not only at elevated mutation rates but also when only a single point mutation is present in each of the components, we utilized the previously described 150 random system double mutants (*Figure 2—figure supplement 2*), which consist of a unique combination of a single point mutation in *cis* and a single point mutation in *trans* (*Figure 3—figure supplement 1*; *Figure 3—source data 1*). In a plate reader, and hence at the level of a monoclonal population, we measured expression levels of all 150 system double mutants, as well as their corresponding single mutants, in the presence of CI (*Figure 2—figure supplement 2*). From these measurements, we calculated epistasis as the deviation from the additive expectation based on wildtype-normalized single mutant effects. This definition of epistasis mirrors the convolution approach utilized for the analysis of DMEs in *Figure 2*.

Intermolecular epistasis was common between single point mutations in our system, as 71 of 150 double mutants significantly deviated from their additive expectations (*Figure 3*, *Figure 3—figure supplement 2*). As such, intermolecular epistasis impacts expression levels in the system not only when a relatively large number of mutations accumulate, as shown in *Figure 2*, but also when a single point mutation is introduced in each of the components. Furthermore, 53 of these 71 mutants

**Table 1.** Additive null predictions for interactions between mutations.

Most interactions result in high expression phenotypes because the wildtype in the presence of CI has no expression, meaning that mutations are either neutral or increase wildtype expression. If an effect of a mutation is positive, the additive model states that the effect remains positive (and the same), independent of the genetic background. As such, these predictions are true only for a system that is tightly repressed, where the wildtype has no expression. 'High +' indicates predictions that result in expression levels above the biologically meaningful limit, which is defined by the unrepressed $P_R$ promoter (shown in *Figure 2A*). We treat these predictions as high expression phenotypes. We consider three categorical single component effects ('no expression', 'intermediate expression', and 'high expression'), and show the categorical effect predicted by the additive null model for the system. We use categorical effects only to provide an intuition for what the additive model predicts - to obtain actual predictions of system DMEs, we use convolution (as explained in detail in Materials and methods section *Naïve convolution of component distributions as the null model for additivity between mutations*).

|  |  | Effects of mutations in *trans* | | |
|  |  | No expression | Intermediate expression | High expression |
| --- | --- | --- | --- | --- |
| **Effects of mutations in *cis*** | **No expression** | No | Intermediate | High |
|  | **Intermediate expression** | Intermediate | Intermediate +high | High + |
|  | **High expression** | High | High + | High + |

DOI: https://doi.org/10.7554/eLife.28921.015

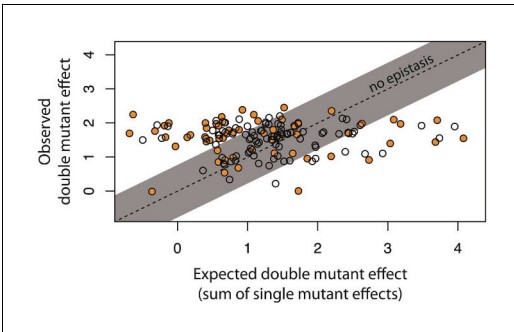

**Figure 3.** Epistasis in 150 random system double mutants. We created 150 double mutants with one unique random point mutation in the *cis*- and the other in the *trans*-element (*Figure 3—figure supplement 1*; *Figure 3—source data 1*). In a plate reader, we measured expression levels of monoclonal populations of each double mutant and its constitutive single mutants in the presence of CI. Epistasis was estimated as the deviation of the observed double mutant effect from the additive expectation based on single mutant effects (shown for each double mutant in *Figure 3—figure supplement 2*). The grey bar indicates measurements that are not in significant epistasis. Double mutants above the 'no epistasis' line are in positive epistasis (observed double mutant effect is greater than the additive expectation), and those below are in negative epistasis. Distributions of mutational effects for single *cis* and *trans* mutants, as well as for the double system mutants are shown in *Figure 2—figure supplement 2*, while the data for epistasis calculations are shown in *Figure 3—source data 2*. By measuring expression levels of 60 random double mutants (shown in orange) and their constitutive single mutants in the flow cytometer, we confirmed that estimates of epistasis using the plate reader correspond to those based on flow cytometry measurements (*Figure 3—figure supplement 6*; Source Data 3), and that the noise of expression is constant for all mutants (*Figure 3—figure supplements 3*, *4* and *5*; Source Data 4).
DOI: https://doi.org/10.7554/eLife.28921.016

The following source data and figure supplements are available for figure 3:

**Source data 1.** Identity and location of mutations in the 150 random double mutant library.
DOI: https://doi.org/10.7554/eLife.28921.023

**Source data 2.** Calculating epistasis from the effects of 150 random double mutants and their corresponding single point mutations, measured in plate reader.
DOI: https://doi.org/10.7554/eLife.28921.024

**Source data 3.** Gene expression noise in single and double mutants is constant.
DOI: https://doi.org/10.7554/eLife.28921.025

**Source data 4.** Calculating epistasis from the effects of 60 random double mutants and their corresponding single point mutations, measured in flow cytometer.
DOI: https://doi.org/10.7554/eLife.28921.026

*Figure 3 continued on next page*

were in positive epistasis, meaning that the double mutant effect was higher than expected based on single mutant effects. Such positive epistasis contributes to the observed increase in the frequency of mutants with intermediate phenotypes in the system (*Figure 2—figure supplement 2*), as seemingly neutral *trans* mutations, which fully repress the wildtype *cis*, show elevated expression on mutated *cis* backgrounds. On the other hand, the presence of negative epistasis indicates a penetrant *trans* mutation, whose high expression on the wildtype *cis* is decreased on a mutated *cis* background. Such epistasis most frequently arises from loss-of-function *trans* mutations, in which the system expression level in the presence of CI is determined only by the effect of the *cis* mutation in the absence of CI. Consequently, we observed negative epistasis in 8 of 10 *trans* single point mutants that exhibited no measurable repression (*Figure 3—source data 2*). We did not identify any relationship between the presence of intermolecular epistasis and the physical location of mutations in the *trans*- ($\chi^2_{(4)}=2.02$; p=0.73) or the *cis*-element ($\chi^2_{(5)}=1.69$; p=0.89)(*Figure 3—source data 1*), indicating that even though individually mutations in some loci have a greater effect on expression level, they are not associated with any particular form of epistasis.

## Intermolecular epistasis arises from the genetic regulatory structure of the system

In the system we study, the genetic regulatory structure (*Figure 1*) indicates that mutations in cis affect both the binding of RNAP and of the repressor. A comparison to the 'naïve' convolutions performed without accounting for this regulatory structure (*Figure 2*) demonstrates that the presence of intermolecular epistasis prevents accurate predictions of system DME from individual component DMEs. We wanted to understand if these predictions could be improved by accounting for the effects of *cis*-element mutations on RNAP binding, which are measured in the absence of CI. In other words, we wanted to connect the basic knowledge of the regulatory structure of the system to the epistasis between mutations in the two components.

To do so experimentally, we combined the high mutation probability *cis* and the low mutation probability *trans*-element libraries (*Figure 4*). We chose these two libraries because they have similar expected number of mutations ($n \sim 6$), therefore minimizing a potential bias in their mutational effects arising from the difference in

*Figure 3 continued*

**Figure supplement 1.** Identity and location of mutations in the 150 random double mutant library.
DOI: https://doi.org/10.7554/eLife.28921.017

**Figure supplement 2.** Single mutant effects, as well as predicted and observed double mutant effects.
DOI: https://doi.org/10.7554/eLife.28921.018

**Figure supplement 3.** 30 double mutants with their corresponding single mutants, which are in significant positive epistasis.
DOI: https://doi.org/10.7554/eLife.28921.019

**Figure supplement 4.** Ten double mutants with their corresponding single mutants, which are in significant negative epistasis.
DOI: https://doi.org/10.7554/eLife.28921.020

**Figure supplement 5.** Twenty double mutants with their corresponding single mutants, which are not in significant epistasis.
DOI: https://doi.org/10.7554/eLife.28921.021

**Figure supplement 6.** Flow cytometer and plate reader measurements give equivalent estimates of epistasis.
DOI: https://doi.org/10.7554/eLife.28921.022

the number of mutations in the two components. We used FACS to partition the mutant libraries into the three phenotypic bins (no expression, intermediate, and high expression, as in *Figure 2*). In this manner, we partitioned two libraries: the high mutation *cis*-element library in the absence of CI (*Figure 2—figure supplement 1D*) and the low mutation *trans*-element library in the presence of CI (*Figure 2E*). We constructed nine new mutant libraries with all possible combinations of these partitions (*Figure 5A–I*). From these combination DMEs, measured in the presence of CI, we calculated the expected frequencies of mutants in each of the three categories, by weighting each combination DME by the relative frequency of the original *trans*-element partition it was derived from. Then, the predicted frequency in each category of the system DME is the sum of the weighted counts in the corresponding category across all nine DMEs (see Materials and methods). These predicted frequencies were not different from the observed ones (*Table 2*). As such, only by experimentally accounting for the genetic regulatory structure of the system (*Figure 1*) can we accurately predict mutant frequencies in three biologically meaningful categories and qualitatively explain how the *cis* background alters the phenotypic effects of *trans*-element mutations, as detailed in the legend to *Figure 5*.

We also produced a mathematical prediction of the system DME that incorporated the knowledge of its genetic regulatory structure. To do so, in addition to incorporating the knowledge of the effects of *cis* mutations in the absence of CI, we also assumed that all *trans*-element mutations that have high expression (same expression as the wildtype in the absence of CI) are loss-of-function mutants, which do not bind any mutated *cis*. When convolving the two component distributions, the effects of such loss-of-function *trans* mutants are removed and replaced by the *cis*-element DME in the absence of CI. This approach, which accounts for the effects of *cis* mutations on RNAP binding, captures the frequencies of mutants in the three phenotypic categories ('no', 'intermediate', and 'high' expression) more accurately than the 'naïve' convolutions (*Figure 4*; *Table 2*; *Figure 2—figure supplement 6*). From a theoretical evolutionary perspective, it is the deviations from a simple additive model that have well-documented consequences for organismal evolution, as they determine the ruggedness of the adaptive landscape (*de Visser and Krug, 2014*). Here, we show how those deviations emerge from the underlying genetic regulatory structure, and hence how they might lead to better predictions of regulatory system DMEs.

## Accounting for the genetic regulatory structure of the system does not explain all intermolecular epistasis

While considering the effects of *cis*-element mutations on RNAP binding explains intermolecular epistasis to an extent that allows more accurate predictions of system DMEs, it might not explain all of it. To determine the extent to which intermolecular epistasis cannot be explained by accounting for the genetic regulatory structure of the system, we constructed a second library of 150 system double mutants. This time, instead of randomly combining point mutations in *cis*- and *trans*-elements, we combined point mutations with a specific phenotype. Namely, all *cis*-element point mutants exhibited high expression in the absence of CI, meaning that the binding of RNAP was not measurably impaired (*Figure 6—figure supplement 1A*). All point mutations in trans used to assemble the 150 double mutants exhibited no expression (*Figure 6—figure supplement 1B*), meaning that these mutants had a fully functional *trans*-element on the wildtype *cis* background. The system double mutant library made in this manner corresponds to the partition combination shown in

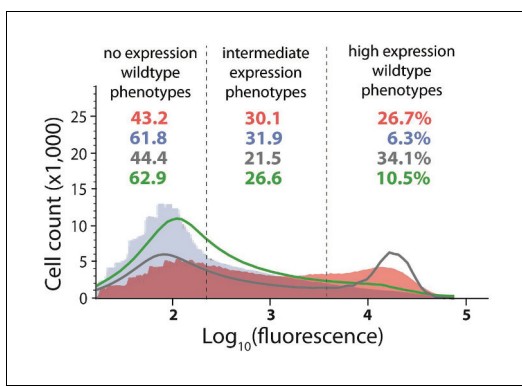

**Figure 4.** Distribution of expression phenotypes for the system with high mutation probability in the *cis*-element and low mutation probability in the *trans*-element. The distributions of fluorescence are shown in the absence of CI (shown in red) and in the presence of CI (shown in blue). Each distribution was obtained by pooling two independent measurements of 500,000 cells. The dashed lines separate three categories of phenotypes – no expression phenotypes (corresponding to repressed wildtype); intermediate expression phenotypes; and high expression phenotypes (corresponding to the wildtype in the absence of CI). Numbers are percentage of counts in each category, in the absence (red) and in the presence (blue) of CI. The naïve convolution-predicted DME in the presence of CI, performed in the absence of any knowledge of the genetic regulatory structure of the system, is shown in grey, together with the corresponding frequencies of mutants in each category. The convolution prediction that accounted for the regulatory structure is shown in dark green. Results of the Pearson's Chi-squared test for the differences between the observed and both types of convolution-predicted DMEs in the presence of CI are shown in *Table 2*.

DOI: https://doi.org/10.7554/eLife.28921.028

*Figure 5C*. In such a library, in the presence of CI, the additive null model predicts that a double mutant would exhibit the same phenotype as its *cis* point mutant, when the corresponding *trans* mutant maintains the same binding properties as the wildtype CI. When this is true, the system double mutant would not be in significant epistasis. Conversely, the system double mutant in this library would be in significant epistasis only when the *trans* mutant binds the mutated *cis* differently than the wildtype *trans* does.

We found that 15 double mutants in this library are in significant epistasis (*Figure 6*). These mutants maintained a decreased yet substantial dynamical range between the two environments (absence and presence of CI), and hence were still functional regulators (*Figure 6—source data 1*). Furthermore, all 15 were in positive epistasis, indicating that the double mutant effect is greater than the additive expectation. In such mutants, *trans* mutations are phenotypically neutral on the wildtype *cis*, but not on a mutated *cis* background, meaning that the *trans* mutant binds the mutated *cis* less strongly than the wildtype CI does. It is worth noting that the lack of mutants that are in negative epistasis in this library might be due to the strong wildtype binding of the CI repressor, so that introducing point mutations in trans that improve binding is highly unlikely. When mutations in trans induce alterations to the binding properties of the repressor (but not complete loss of function), intermolecular epistasis cannot be accounted for by the underlying structure of the genetic regulatory network. Interestingly, a disproportionate number of double mutants that were in positive epistasis carried a *trans* mutation in the linker region of the CI ($\chi^2_{(4)}=20.66$; $p<0.0005$), which connects the N-terminal DNA-binding domain with the C-terminal dimerization domain (*Figure 6—figure supplement 2*; *Figure 6—source data 2*). This is in contrast to the random double mutant library (*Figure 3*), where we found no relationships between location of mutation and epistasis.

## Discussion

In this study, we show that mutating a molecular system with the most common transcriptional regulatory structure in prokaryotes (*Salgado et al., 2013*), namely a repressible promoter, increases phenotypic variation beyond what can be achieved by mutating any of the individual components alone. We focused on the phenotypic effects, rather than the fitness effects of mutations, in order to minimize the complexity of the studied system and also to enable a more direct interpretation of the results, as fitness effects of mutations depend on a much larger, and often unknown, set of factors than simply their phenotypic effects. Doing so enables an interpretation of observed DMEs in the light of their underlying molecular mechanisms, as recently shown for a prokaryotic (*Lagator et al., 2017*) and a eukaryotic *cis*-regulatory element (*White et al., 2016*). Furthermore, while previous studies investigated the effects of specific mutations in the contact surface between two molecules (*Anderson et al., 2015*; *Podgornaia and Laub, 2015*), their local nature, imposed by the large

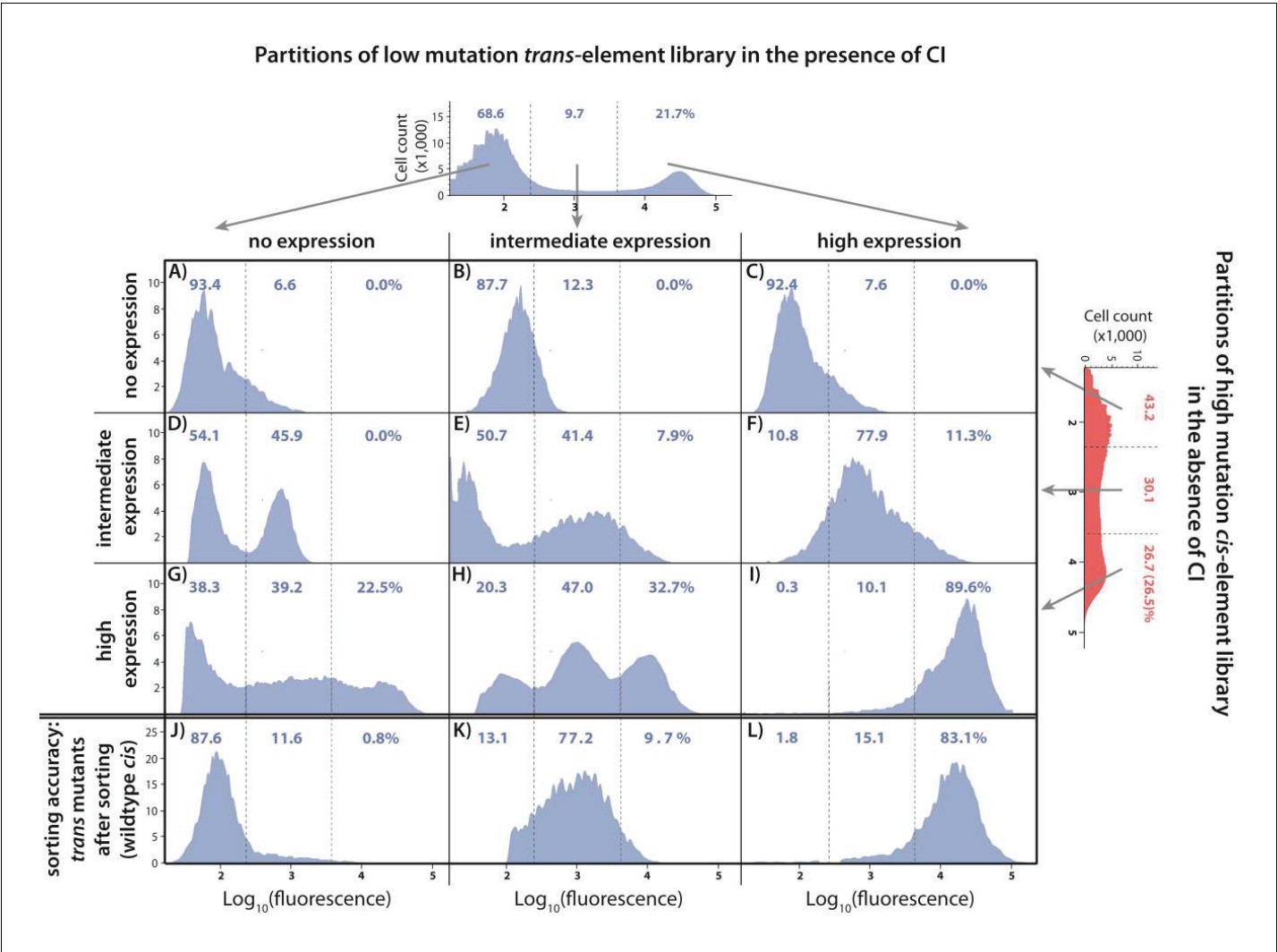

**Figure 5.** Understanding the interactions between mutations in *cis* and *trans* by accounting for the genetic regulatory structure of the system. Three partitions, obtained by FACS, of the low mutation probability *trans*-element library (corresponding to no expression, intermediate, and high expression phenotypes) were combined with the three equivalent partitions of the high mutation probability *cis*-element library in the absence of CI. The *cis*-element library in the absence of CI shows only the effect of mutations on RNAP binding. The DME of the original *trans*-element library, corresponding to *Figure 2E*, is shown on top. The DME of the original *cis*-element library, corresponding to *Figure 2—figure supplement 1D*, is shown on the right. The arrows illustrate from which category of the original DME (either *trans* or *cis*) were the sorted mutants used to make the particular combined library. DMEs of all nine partition-combination libraries were estimated using flow cytometry, with each distribution obtained by pooling two independent measurements of 500,000 cells. Also shown is the sorting accuracy, obtained as the repeated DME measurement of each *trans* partition following the original FACS sorting (panels J,K,L). The distributions of fluorescence are shown in the presence of CI. The dashed lines separate three categories of phenotypes – 'no expression', 'intermediate', and 'high expression' phenotypes. Numbers are percentage of counts in each category. At least in part, intermolecular epistasis can be qualitatively explained by considering the genetic regulatory structure of the system, as follows. Panels A), (**B**), and C): no expression *cis*-element mutants do not bind RNAP sufficiently to lead to expression, so that system mutants containing them remain in the 'no expression' bin irrespective of the effect of mutations in trans. (**D**) No expression *trans*-element mutants fully repress on wildtype *cis*. When *cis*-element mutations of intermediate expression are introduced, some still bind the functional CI mutants leading to repression, while others carry mutations that prevent CI binding, resulting in intermediate expression system mutants. (**E**) Intermediate expression CI mutants only partially repress on wildtype *cis*, but fully repress some *cis*-element mutants that have lowered RNAP binding. Other intermediate expression CI mutants do not bind a mutated *cis*-element background. (**F**) High expression CI mutants cannot bind wildtype *cis*. Similarly, they do not bind intermediate expression *cis*-element mutants, resulting in intermediate expression in the system. (**G**), (**H**): Some high expression *cis*-element mutants can fully bind mutated but functional CI mutants, others can only partially bind them, while some maintain full RNAP binding while losing all CI binding. (**I**) non-functional CI mutants do not repress on wildtype *cis*, hence also not on mutated *cis*-element backgrounds.

DOI: https://doi.org/10.7554/eLife.28921.029

**Table 2.** Predicting the system DME is possible only when accounting for epistasis between components.

Four different predictions for the frequency of mutants in each partition (no expression, intermediate, and high expression phenotypes) were compared to the actual system distribution, measured in the presence of CI, and consisting of the high probability *cis* +low probability *trans* libraries (*Figure 4*). The experimental prediction based on distributions of nine partition-combination DMEs (*Figure 5*) was obtained by weighting the nine DMEs with the relative frequency of the original *trans* partition they were derived from (from *Figure 2E* - no expression partitions were weighted by 0.686; novel phenotype partitions by 0.097; high expression partitions by 0.217). One convolution of *cis*- and *trans*-element DMEs was performed in the absence of any knowledge of the genetic regulatory structure of the system (naïve convolution), and the other by accounting for the effects of *cis* mutations in the absence of CI (*Figure 2—figure supplement 7*). The null prediction tested if the observed distributions could arise only from the imprecise sorting of the original *trans* library partitions (*Figure 4J,K,L*). Only the experimental prediction based on partition-combination libraries and the convolution that accounted for the genetic regulatory structure could explain the epistatic interactions between mutants in *cis*- and *trans*-elements (shown in orange).

| | Actual system (mutations in cis and trans) | Experimental prediction (based on partition libraries) | Convolution (no knowledge of genetic structure) | Convolution (accounting for cis effects in the absence of CI) | Null prediction (based on sorting accuracy) |
|---|---|---|---|---|---|
| No expression phenotypes (%) | 61.8 | 55.1 | 44.4 | 62.9 | 68.6 |
| Intermediate phenotypes (%) | 31.9 | 31.1 | 21.5 | 26.6 | 9.7 |
| High expression phenotypes (%) | 6.3 | 13.8 | 34.1 | 10.5 | 21.7 |
| | $\chi^2$ compared to actual distribution | 5.49 | 24.7 | 1.89 | 59.8 |
| | p value | 0.064 | $10^{-6}$ | 0.388 | $10^{-13}$ |

DOI: https://doi.org/10.7554/eLife.28921.027

mutational space of proteins, prevented conclusions about DMEs of molecular systems. Precisely because our experimental approach allowed us to overcome these obstacles, we were able to show how the regulatory network structure determines intermolecular epistasis, indicating a broad range of conditions under which the additive null models of interactions between mutations might be inaccurate.

The observed increase in the frequency of intermediate phenotypes arises, in large part, from intermolecular epistasis between the two components of the system, most of which can be attributed to the structure of the gene regulatory system. The observed increase is due to both positive and negative epistasis. When intermolecular epistasis is positive, mutations in *cis* expose the genetic variation hidden in originally phenotypically neutral *trans* mutations. This can be achieved in two ways: (i) through changes in the RNAP and repressor binding sites, which are accounted for by considering the genetic regulatory structure of the system (*Figure 5*); or (ii) when mutations in trans change the binding preference of the repressor, so that a mutation in *cis* decreases the binding of the mutated *trans* more than it decreases the binding of the wildtype repressor (*Figure 6*). Negative intermolecular epistasis, on the other hand, arises when: (i) the *trans* mutations are penetrant, so that their effects (in particular, increased expression) are buffered by mutations in *cis*. This epistasis is frequently observed when *trans* mutations lead to a complete loss of binding (high expression phenotype), so that the system phenotype becomes the same in the presence and in the absence of CI (*Figure 3*). In other words, the system undergoes a qualitative transition from a three-component and thus a regulated promoter, to a two-component or a constitutive promoter. Under these conditions, the system phenotype is determined only by the effect of *cis* mutations on RNAP binding and can be thus explained by considering the genetic regulatory structure of the system (*Figure 5*). (ii) Negative epistasis can also arise when *trans*-element mutations alter the binding preference of the repressor, so that the mutated *trans* binds mutated *cis* more strongly than the wildtype repressor does. We found no evidence for this type of epistasis in the library of 150 random double mutants, likely due to our use of the Lambda $P_R$ promoter, which binds the repressor very tightly.

In the studied system, we demonstrate that intermolecular epistasis is present even between single point mutations in the *cis*- and the *trans*-element (*Figure 3*), and not only when a larger number

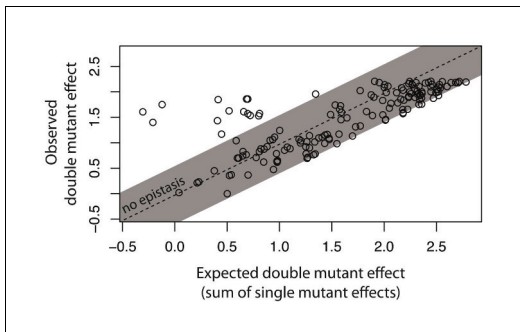

**Figure 6.** Not all intermolecular epistasis can be explained by accounting for the underlying genetic regulatory structure of the system. We created 150 double mutants with single mutation combinations corresponding to *Figure 5G*. A double mutant in this library would not be in epistasis unless a mutation in *trans* binds the *cis* mutant differently than the wildtype *trans* does. In a plate reader, we measured expression levels of monoclonal populations of each double mutant and its constitutive single mutants in the presence of CI. Epistasis was estimated as the deviation of the observed double mutant effect from the additive expectation based on single mutant effects. The grey bar indicates measurements that are not in significant epistasis. The effects of single mutations in *cis* and in *trans*, as well as the double mutant effects, are shown in *Figure 6—figure supplement 1*, while the underlying data are shown in *Figure 6—source data 1*. The location of point mutations is shown in *Figure 6—figure supplement 2* and *Figure 6—source data 2*.

DOI: https://doi.org/10.7554/eLife.28921.030

The following source data and figure supplements are available for figure 6:

**Source data 1.** Calculating epistasis from the effects of 150 double mutants and their corresponding single point mutations, measured in plate reader.

DOI: https://doi.org/10.7554/eLife.28921.033

**Source data 2.** Identity and location of mutations in the library of 150 double mutant, with a point mutation in *cis* that leads to high expression in the absence of CI, and a point mutation in trans that leads to no expression in the presence of CI.

DOI: https://doi.org/10.7554/eLife.28921.034

**Figure supplement 1.** Distribution of single mutation effects in 150 system double mutants and their corresponding single mutants.

DOI: https://doi.org/10.7554/eLife.28921.031

**Figure supplement 2.** Identity and location of mutations in the double mutant library with a *trans* mutation that has no expression in the presence of CI, and a *cis* mutation with high expression in the absence of CI.

DOI: https://doi.org/10.7554/eLife.28921.032

of mutations accumulates in each component (*Figure 2*). When considering interactions between point mutations (*Figure 3*;6), we identified that positive epistasis contributes disproportionately to the increase in the frequency of intermediate phenotypes. The extent to which positive, as opposed to negative, epistasis drives the observed increase in the intermediate phenotypes when a greater number of mutations are present in the components (*Figure 2*) cannot be addressed without knowing the effects of specific molecular interactions between a prohibitively large number of mutation combinations.

Epistasis has been shown to impose constraints on evolutionary paths by increasing the ruggedness of the adaptive landscape both in theoretical and experimental studies (*Whitlock et al., 1995*; *Barton and Partridge, 2000*; *Weinreich et al., 2006*; *Poelwijk et al., 2011*; *Breen et al., 2012*; *Podgornaia and Laub, 2015*). Our data suggest that, for transcriptional regulatory networks, mutating the system of specifically interacting components alleviates the biophysical and mechanistic constraints acting on individual components, and in doing so increases phenotypic variation accessible through mutation. This role of intermolecular epistasis as a facilitator rather than a constraint on evolution arises, in significant part, directly from the genetic structure of a repressible transcriptional regulatory system. As such, it might be a common feature in prokaryotes, where repression through direct binding site overlap with RNAP forms the most common type of transcriptional regulatory system organization (*Salgado et al., 2013*). In these and other similar systems, potential paths for protein evolution might be more abundant when the interacting DNA-binding site is also mutating, as mutations in the partner component can expose the genetic variation hidden in originally phenotypically neutral mutations. Such intermolecular epistasis could give rise to punctuated protein evolution (*Fontana and Schuster, 1998*) – long periods of phenotypic stasis during which a transcription factor accumulates neutral mutations, interrupted by rapid adaptive evolution facilitated by mutations in the DNA-binding site. Furthermore, a gene might be horizontally transferred with its cognate *cis*-element, but without its cognate regulator. Then, intermolecular epistasis between the transferred *cis*-element and the orthologous transcription factor may reveal previously unavailable phenotypes that can facilitate adaptation to new niches. Therefore, accumulating mutations in the entire system, as opposed to only in a single

component, appears to facilitate evolution both by extending the neutral sequence space, and hence increasing diversity through drift (*Lynch and Hagner, 2015*), as well as by increasing the available phenotypic variability.

## Materials and methods

### Gene regulation in the lambda phage switch

The *right* regulatory region of Lambda phage ($O_R$) is responsible for the decision-making between lysis or lysogeny (*Johnson et al., 1981*). The regulatory region consists of two RNA polymerase (RNAP) binding sites - promoters $P_R$ and $P_{RM}$ (not shown), and three CI/Cro transcription factor operators ($O_{R1}$, $O_{R2}$, and $O_{R3}$) (*Figure 1A*). In the wildtype system, the strong promoter $P_R$ leads to expression of the transcription factor Cro. The transcription factor CI represses $P_R$ promoter by direct binding-site competition with RNAP.

### Synthetic system

We used a synthetic system based on the Lambda phage switch, in which we decoupled the *cis-* and *trans*-regulatory elements (*Figure 1B,C*). We removed *cI* and substituted *cro* with *venus-yfp* (*Nagai et al., 2002*) under control of $P_R$ promoter, followed by a T1 terminator sequence. The $O_{R3}$ site was removed in order to remove the $P_{RM}$ promoter. Separated by a terminator sequence and 500 random base pairs, we placed *cI* under the control of $P_{TET}$, an inducible promoter regulated by TetR (*Lutz and Bujard, 1997*), followed by a TL17 terminator sequence. In this way, concentration of CI transcription factor in the cell was under external control, achieved by addition of the inducer anhydrotetracycline (aTc). The entire cassette was inserted into a low-copy number plasmid backbone pZS* carrying a kanamycin resistance gene (*Lutz and Bujard, 1997*). In this system, *cI* constitutes the *trans*-element, while the $P_R$ promoter together with the two CI operator sites $O_{R1}$ and $O_{R2}$ make the *cis*-element.

### Creating the mutant libraries

The library of *cis-* and *trans*-element mutants was created using the GeneMorph II™ random mutagenesis kit (Agilent Technologies, Santa Clara, US). We created three mutant libraries with different average probability of mutations (0.01, 0.04, and 0.07 mutation chance per nucleotide) for both the transcription factor (714 bp) and the $P_R$ *cis*-regulatory element (84 bp). Therefore, the average number of mutations per mutant in the *trans*-element libraries was 7, 28, or 49, while in the *cis*-element it was 1, 3, or 6, respectively. We applied the same likelihood of mutation per nucleotide rather than using the same actual number of mutations between equivalent *cis-* and *trans*-element libraries in order to more accurately represent the biological process of mutagenesis.

PCR products of mutagenesis reactions were ligated into the wildtype construct, and inserted into One-Shot Top10 cells (Life Technologies, Carlsbad). This step was used to maximize the library diversity due to One-Shot Top10 cells' high competency. Following electroporation, cells were plated at low concentrations on selective kanamycin plates to allow single colony formation and minimize resource competition, and grown overnight. Using chilled LB media, colonies were washed off the plates and collected. To ensure large coverage, we cloned mutagenized PCR products until we obtained at least 30,000 individual colonies (uniquely transformed individuals). Due to the stochastic nature of the mutagenesis protocol used (Agilent Technologies), the number of uniquely transformed individuals did not necessarily equal the number of different mutant genotypes, especially at low mutation rates. To illustrate, when the mutation rate was low so that a *cis*-element mutant would have on average only one mutation, some PCR mutagenesis products did not contain any mutations. When mutations were in the *trans*-element, which is ~10-fold longer, almost all PCR mutagenesis products contained several mutations. Using the information provided by the supplier (which we verified by sequencing 40 mutants from each *cis* and *trans* library, see below) on the distributions of the number of mutations (given a mutagenesis rate), we estimated that approximately 34.2%, 3.5%, and 0.2% of the low, intermediate, and high mutation number *cis*-element libraries consisted of wildtype genotypes. The comparative proportion of wildtype genotypes in the *trans*-element libraries was 0.07%, $10^{-13}$%, and $10^{-22}$%.

Populations containing a mixture of mutants with a given number of mutations were used to isolate plasmids, and clone them into the modified MG1655 strain expressing *tetR* gene from a constitutive *PN*$_{25}$ promoter. We showed that the distributions of mutation numbers in 40 isolated individuals from each library conformed to the distributions of mutation numbers provided by the supplier. We did this by comparing the actual distribution of the number of mutations to the Poisson distribution based on the predicted mutation probability, using a Kolmogorov-Smirnov (K-S) test (*Figure 2—source data 2*). We used a power test to determine that the sample size of 40 was sufficient to verify the predicted number of mutations, with power set at 0.80 and desired detectable difference of ±0.5 mutations. Among the 240 sequenced mutants from the six libraries (6 × 40), we found no bias toward a specific type of mutations (transitions vs. transversions), nor did we identify overrepresentation of any particular single point mutation or locus in this dataset. From the sequenced mutants, we could estimate the proportion of re-ligated wildtype plasmid, in which the mutated region was not inserted and instead the wildtype used as the cloning template re-ligated back (this is a common occurrence with any library created through standard restriction-digestion cloning techniques). By observing the frequency of wildtype genotypes in the sequenced *trans* libraries (which, due to the relatively large number of mutations should not contain any wildtype sequences), we estimate that < 5% of each library is re-ligated with the wildtype insert (*Figure 2—source data 2*). As this is a relatively small frequency to estimate precisely from sequencing 120 clones (3 × 40 *trans* library sequences), we do not account for it when considering the proportion of wildtype cells in each library. More importantly, this bias should be the same for all libraries and is therefore unlikely to alter our comparative analyses. Finally, among the 240 sequenced plasmids, we did not observe any that contained neither the mutated nor the wildtype insert, as all sequenced plasmids had an insert from cloning.

To make the whole system libraries, we removed through restriction digestion the mutated *cis*-element from each *cis* library and cloned it into the plasmid library already containing *trans*-element mutants with the corresponding mutation probability. Note that system libraries created in this way would likely have somewhat reduced diversity compared to either *cis* or *trans* libraries, as stochastically some mutants present in the component libraries would not find their way into the combined system library. By our design, this potential reduction in diversity would be equally biased toward *cis* and *trans* mutants, and ought not to be inflated for one component compared to the other.

In this study, we set out to understand phenotypic effects of mutations and to connect the DMEs to the epistasis between the two components. To achieve this goal, we investigated the DMEs in random mutant libraries, therefore without knowing the identity of the specific mutants. We did not characterize individual mutants, as drawing conclusion about the effects of specific mutations would require an unachievable high number of mutants to be analyzed. This was because the relatively large number of mutations introduced, in particular in the *trans*-element where each individual in the low mutation probability library contained on average seven point mutations, meant that we covered a very large mutational space. As such, we would need to characterize an astronomical number of mutants to gain the statistical power necessary to discern the effects of individual mutations from the interactions between them. Marginal sampling of such a huge sequence space, which is the best we could achieve, would tell us nothing about how individual positions affect the overall DME.

## Single-cell fluorescence measurements

In order to obtain the distributions of phenotypic effects of mutations in mutant libraries, we used flow cytometry/fluorescence activated cell sorting (FACS) to analyze expression levels of a yellow fluorescence protein. For all libraries, we measured gene expression levels both in the absence and in the presence of the transcription factor CI, determined by absence or presence of the inducer aTc, respectively. Throughout, we use log$_{10}$ of expression level as our phenotype of interest. Mutant libraries, as well as the wildtype construct, were grown overnight in M9 minimal media supplemented by 1% casamino acids, 2% glucose, and 30 μg/ml kanamycin, and either without or with 8 ng/ml aTc. From this point, the investigator was blinded with respect to the identity of each processed library. Overnight populations were diluted 100 times, grown for 2 hr, diluted a further 10 times and their fluorescence measured in a BD FACSAria$^{tm}$ III cell sorter. Fluorescence of 500,000 cells was measured for each replicate of each library. Two independent replicates of each mutant library and the monoclonal wildtype population in each of the two growth conditions were measured in the manner described above. All flow cytometry data were subsequently analyzed in FlowJo

version 10.0.8r1, and measurements with extreme FSC-A and SSC-A values were excluded from the analyses. The two replicate measurements of each library exhibited the same distributions of fluorescence phenotypes (tested by the K-S test) and were pooled together, to give a million counts for each library.

We verified that 1 million individual measurements, as well as the library diversity of at least 30,000 mutants, accurately described phenotypic distributions of possible mutations. To ensure that we are capturing a significant proportion of all possible phenotypic effects, we subsampled progressively smaller number of measurements (n = 500,000; n = 250,000; n = 100,000; n = 50,000) of *cis*- and *trans*-element mutant libraries, in each relevant environment (both absence and presence of CI for *cis*-element libraries, and only presence of CI for *trans*-element libraries). We quantified the difference between each subsampled dataset and the corresponding full dataset using a K-S test, and found that, by randomly subsampling each dataset 50 times, the distributions of phenotypes in subsamples were not statistically different from the distribution of the full dataset (*Figure 2—source data 4*).

## Calculating entropy of a DME

Shannon entropy was used as an estimate of how uniform the distribution is across the whole range of possible phenotypes. The range of possible phenotypes was defined by the minimum and the maximum fluorescence measurement in the entire dataset, across all measured mutant library DMEs. Entropy was calculated as:

$$S = -\sum_k P_k \log P_k + \log \Delta x$$

where $P_k$ is the frequency of fluorescence measurements in the $k^{th}$ bin, and $\Delta x$ is the width of the bin, which was set to 0.05. In principle, values of entropy estimates depend on the bin width, so we checked explicitly that our conclusions do not depend on this particular choice. Error associated with each entropy measurement was calculated using standard bootstrapping methods. We performed a nonparametric permutation test to assess if the differences in entropy are significant.

## Estimating gene expression noise from flow cytometry measurements

We randomly isolated 20 *cis*, 20 *trans*, and 20 system mutants, from each mutation probability library, giving rise to 180 isolates. Power analysis (power.anova.test function in R statistical package) indicated that 20 samples in each category were sufficient to detect differences in gene expression noise of 2%, at significance level of 0.05 and power greater than 0.9. In a flow cytometer, we measured gene expression levels in the presence of CI in 100,000 individual counts, for two replicates of each mutant isolate, as well as for the wildtype. First, using a K-S test, we confirmed that the two replicate distributions for each mutant were not significantly different. We combined the two replicate measurements, and then randomly sampled without replacement 5000 reads from this common pool ten times (*Figure 2—figure supplement 3*, *4* and *5*). Gene expression noise for each such subsample of 5000 reads was estimated as the coefficient of variation, as done in other studies on gene expression noise (*Metzger et al., 2015*). Note that the gene expression noise measured in this manner comes from two sources – the heterogeneity of gene expression between individual cells and the measurement error inherent in the flow cytometer. We assume that the measurement error is constant between all mutants, so that possible changes in the coefficient of variation would indicate a difference in the heterogeneity of gene expression between genetically identical individuals. Using ANOVA (aov function in R statistical software, version 3.4.1), we asked if there were differences in the noise of gene expression between the 60 mutants of the same mutation probability, as well as between all 180 tested mutants. We performed this test separately for the mutants that had no expression (that were fully contained in the 'no expression' category), and for all other mutants. These two groups were treated separately because the flow cytometry fluorescence measurement is not responding to the same intracellular environment when the cell is producing a fluorescence protein and when it is not. As such, estimates of gene expression noise in the two categorically different types of intercellular environments are not directly comparable (*Figure 2—source data 1*). Note that the tests carried out across all three mutation probabilities, which included 95 mutants with no fluorescence and 85 mutants with fluorescence, found no differences in gene expression noise. The probability of mutants with significantly different noise existing in the library but not being detected

at such sample sizes is less than $10^{-4}$ (calculated using experimentally-observed within and between group variance and at power level of 0.9). For a library of 30,000 random mutants, this would mean that no more than 10 system mutants could have different gene expression noise. For such a small number of mutants to skew the observed system DMEs and lead to an increase in intermediate phenotypes would be possible only if they were strongly overrepresented in the library. This is unlikely the case, since all 180 isolated mutants had growth rates that were comparable to the wildtype, and hence increase in noise would have to be hugely beneficial on a very short time scale (around 10 generations).

## Naïve convolution of component distributions as the null model for additivity between mutations

Let us consider the effects of mutations, $m_{cis}$ and $m_{trans}$, in two components of a system. As our phenotype of interest is $\log_{10}$ of fluorescence level, we assume that, if two mutations are independent, their effects are additive: $m_{expected} = m_{cis} + m_{trans}$. A deviation from this additive assumption is termed epistasis, so that: $\varepsilon = m_{observed}$ $m_{expected}$. Note that a deviation from an additive expectation in log is equivalent to a multiplicative prediction on the linear scale. When considering a library of mutants in two components, their effects are represented by corresponding DMEs, $f_{cis}(m)$ and $f_{trans}(m)$, where m is the 'true' effect of a random mutation on the wildtype, in the absence of experimental noise and measurement errors. If one could obtain 'true' DMEs for mutant libraries (in the absence of any type of noise or error), then the additive null expectation should follow a simple convolution of the two distributions: $f_{expected} = f_{cis} * f_{trans}$. Any deviation of the observed combined library DME ($f_{observed}$) from the $f_{expected}$ is indicative of epistasis.

In a realistic setting, experimental noise and instrumental error prevent direct measurements of the 'true' underlying DME ($f$), so that any measured DME ($F$) incorporates all the errors with its 'true' DME $f$. This means that, if the experimental noise and instrumental error do not change between mutants and across the dynamical measurement range, as is the case for our data (*Figure 2—source data 1*; *Figure 3—source data 3*), then the observed DME ($F$) of a mutant library is equivalent to a convolution between its 'true' underlying DME ($f$) and the measured wildtype distribution ($F_{wt}$): $F = f * F_{wt}$. In other words, the 'true' DME ($f$) shows how mutations alter wildtype expression. It follows that the additive prediction for the combined library, in the absence of epistasis, is a convolution of three DMEs:

$$F^{+}_{expected} = f^{+}_{expected} * F^{+}_{wt} = f^{+}_{cis} * f^{+}_{trans} * F^{+}_{wt}$$

where $f^{+}_{cis}$ and $f^{+}_{trans}$ are the 'true' distributions of the *cis*- and the *trans*-element, respectively, and the superscript '+' indicates the presence of CI. The same equality would of course hold true in the absence of CI, but the analysis is trivial, as then the *trans* library shows no difference to the corresponding wildtype $F^{-}_{wt}$, that is $f^{-}_{trans}$ is a unit element for the operation of convolution (delta-distribution). For simplicity, we will omit the subscript when discussing DMEs obtained in the presence of CI.

The additive prediction (multiplicative in $\log_{10}$ of expression) can be rewritten in two equivalent forms:

1. $F_{expected} = F_{cis} * f_{trans}$
2. $F_{expected} = f_{cis} * F_{trans}$

To obtain either of the underlying 'true' DMEs, we would need to deconvolve the wildtype distribution from one of the measured component library DMEs. However, although well understood and well behaved analytically, (de)convolutions are known to be highly unstable when used on numerical datasets, like ours. Therefore, we would need at least one of the component distributions in their analytical form. Instead of fitting one of the measured DMEs to some analytical representation followed by a deconvolution, we decided to directly 'reverse engineer' one of the component DMEs. We chose the *cis* DME in the presence of CI, as the simpler of the two. Concretely, we searched for $f_{cis}$, such that its convolution with $F_{wt}$ matches the observed $F_{cis}$ as closely as possible. We assumed $f_{cis}$ to be from the gamma-family, as a relatively wide family of curves often used to describe DMEs (*Figure 2—figure supplement 7A*). We optimized three parameters of the $f_{cis}$: shape, scale, and location, to minimize the squared differences between the observed *cis*-element DME ($F_{cis}$) and $f_{cis} * F_{wt}$ (*Figure 2—figure supplement 7B*). Note that the 'true' *cis* DME indicates that effectively all mutations in *cis* are either neutral or they increase expression. Because of this, convolving the 'true'

*cis* distribution with any other distribution results in an overall increased expression, independently of where the original distribution is centered. This is in line with the usual treatment of single mutant effects: if an effect of a mutation is positive, the additive model states that the effect remains positive (and the same), independently of the genetic background.

After we obtained the 'true' DME for *cis* ($f_{cis}$), we convolve it with the observed *trans* DME to produce a naïve null-model for the system DME in the absence of epistasis. Convolving without any adjustments results in a part of the predicted system DME that lies beyond the highest experimentally recorded fluorescence levels (*Figure 2—figure supplement 7C*). Because we use one of the strongest known promoters, the Lambda $P_R$, predictions that have higher expression than the wildtype are not biologically meaningful, as no combination of component mutations could experimentally result in such high expression levels. To reflect this maximal biologically obtainable limit to expression levels, we introduce a cutoff, effectively treating any mutant that would result in higher expression levels as having the wildtype expression. In practice, we (i) removed the high expression peak from the *trans* DME (as convolution with those mutants gives rise to higher expression levels), (ii) performed a convolution between the remainder of the *trans* DME and the $f_{cis}$, (iii) introduced a smooth cutoff at maximum expression levels to the convolved distribution, and (iv) added back the residual part of the high-expressing mutants. More specifically in the first step, we fitted the fraction $\alpha$ of the wildtype distribution in the absence of CI ($F^-_{wt}$) to minimize its (square) difference to the right-hand part of the high-expression *trans* peak (*Figure 2—figure supplement 7D,E*). In this way, we obtained a smooth remainder to convolve with $f_{cis}$, which will produce biologically realistic values. In the end, we add back the distribution of the wildtype in the absence of CI ($F^-_{wt}$), so to obtain a properly normalized predicted distribution (*Figure 2—figure supplement 7F*). Because we impose this limit to the highest biologically obtainable expression level, convolving a hypothetical distribution consisting of predominantly high expression phenotypes with the 'true' *cis* distribution ($f_{cis}$) would result in only high expression phenotypes.

The prediction for the system DME obtained in this manner reflects only two assumptions: the additive assumption of no epistasis, and the limit to maximal attainable expression levels. As such, this naïve prediction explicitly disregards any information about the genetic regulatory structure of the system. For each mutation probability, we evaluate if the predicted DME is different from the experimentally observed DME by conducting Pearson's Chi-squared test to compare the frequencies of mutants in the three expression categories ('no', 'intermediate', and 'high' expression).

## Epistasis between random point mutants

We created 150 mutants with a random point mutation in *cis*, and 150 mutants with a random point mutation in trans. *Cis*-element mutants were identified by Sanger sequencing of 400 randomly selected mutants from the *cis*-element library with low (0.01) mutation probability. To obtain 150 *trans*-element mutants, we repeated the random mutagenesis protocol on *cI* with a very low mutation rate (yielding approximately one mutation per kb). From this reaction, we randomly selected and sequenced 500 mutants in order to identify 150 that contained only a single point mutation. Then, we created a library of 150 double mutants, with one point mutation in the *cis*- and the other in the *trans*-element. These 150 double mutants were unique, as each one consisted of a unique pairing between *cis* and *trans* point mutations, so that no point mutations were found in more than one double mutant (*Figure 3—source data 1*).

In a plate reader, we measured fluorescence levels of all 150 double mutants as well as of their corresponding point mutants. The mutants, as well as the wildtype, were grown overnight in M9 minimal media supplemented with casamino acids, 30 µg/ml kanamycin, and either in the absence of CI or in the presence of CI (induced with 8 ng/ml of aTc). The overnight populations were diluted 1000-fold, grown until $OD_{600}$ of approximately 0.05, and their fluorescence measured in a Bio-Tek Synergy H1 platereader. This procedure was replicated six times for each mutant. We performed a series of pairwise t-tests in order to determine which isolates had significantly different fluorescence to the wildtype. Using a K-S test, we compared if the system double mutants had a higher frequency of intermediate phenotypes.

Consistent with convolution-based analyses, we consider expression level as the $log_{10}$ of fluorescence, so that epistasis is defined as a deviation from an additive model, as $\varepsilon = m_{system} - (m_{cis} + m_{trans})$, where $m_{system}$ is the wildtype-relative fluorescence of a system double mutant, and $m_{cis}$ and $m_{trans}$ the wildtype-relative fluorescence of the two corresponding single mutants. Epistasis

was calculated in the presence of CI, as in the absence of CI all *trans* mutants exhibited wildtype expression, and all system double mutants had the same expression as the *cis* mutant alone. In order to statistically determine which double mutants exhibited epistasis (i.e. ε not equal 1), we conducted a series of FDR-corrected *t*-tests. The errors were calculated based on six replicates, using error propagation to account for the variance due to normalization by the wildtype. Variance was not significantly different between measured mutants (*Figure 3—source data 2*).

It is possible that the estimates of epistasis through population-level measures of fluorescence levels in a plate reader might not be equivalent to estimates obtained through flow cytometry. This would particularly be true if the gene expression noise varied significantly between mutants. In order to confirm this is not the case in our study, we randomly selected 30 double mutants that, based on plate reader measurements, were in significant positive epistasis, 10 double mutants that were in significant negative epistasis, and 20 mutants that were not in significant epistasis. Then, we measured the fluorescence in 100,000 individual reads for two replicates of each isolate in a flow cytometer, in the absence and in the presence of CI (*Figure 3—figure supplement 3*, *4* and *5*). First, we compared if gene expression noise between single and double mutants was the same, by conducing the same kind of analysis as described above, and found no differences between mutants (mutants with no expression: $F_{83,747} = 0.891$; p=0.59; mutants with expression: $F_{95,855} = 1.332$; p=0.174) (*Figure 3—source data 3*). We also confirmed that the noise in single/double mutants was not different to the gene expression noise in isolates from low, intermediate, and high mutation probability libraries (mutants with no expression: $F_{166,1494} = 0.765$; p=0.746; mutants with expression: $F_{180,1620} = 1.385$; p=0.485). Then, we calculated epistasis in the presence of CI from flow cytometry measurements in the same manner as described for plate reader measurements (*Figure 3—source data 4*). To evaluate the significance of calculated deviations from the additive expectation (epistasis), we use error propagation on the standard deviation obtained from the combined flow cytometry distributions of the two replicates, and not on the variance between means of replicate measurements (since the measured means for each isolated mutant were near identical between replicates). Linear regression between the estimates of epistasis from the two types of measurements shows that flow cytometry gives the same description of epistasis as the plate reader measurements ($F_{1,58} = 350.5$; p<0.0001) (*Figure 3—figure supplement 6*).

Because all 150 double mutants were sequenced, we could test if epistasis was associated with the location of mutations. For the *trans*-element, we identified three locations: the N-terminus and the C-terminus domains, and the linker region between them (*Figure 3—source data 1*). For the *cis*-element, the mutations could either be in one of the CI operator sites, in the RNAP contact residues (−10 and −35 regions), in the sites that have direct contact with both, or those that do not have direct contact with either protein (*Figure 3—source data 1*). Then, we tested if existence of epistasis depended on the location of point mutations through a Pearson's Chi-squared test, which considered only the binary value for epistasis: either the presence of significant epistasis or its absence.

## Experimentally accounting for the genetic regulatory structure of the system

We wanted to explore if accounting for the genetic regulatory structure of the studied system would improve our ability to predict the system DME from the DMEs of its components. To this end, we put together the low mutation probability *trans*-element (*Figure 2E*) and high mutation probability *cis*-element libraries (*Figure 2—figure supplement 1D*) and measured the DME for this library in the manner described above. We put together these two libraries as they have approximately the same average number of mutations (seven for the *trans*- and six for the *cis*-element), allowing a comparison that is not influenced by the actual number of mutations in each of the two elements.

In order to experimentally predict the frequencies of phenotypes in each category of the low mutation *trans*-element +high mutation *cis*-element library, we partitioned the low mutation probability *trans*-element library in the presence, and high mutation probability *cis*-element library in the absence of CI. Using FACS, we partitioned each library into three bins, corresponding to the no expression, intermediate, and high expression phenotype categories. We sorted a minimum of 500,000 individuals into each bin, and grew them overnight in LB with 30 μg/ml kanamycin. Using these populations, we obtained a measure of sorting accuracy by obtaining a DME of each *trans*-element partition after overnight growth. We isolated the plasmids from all six partitioned populations (three *cis* and three *trans*), and cloned all possible combinations of *cis*- and *trans*-element partitions

to make nine new mutant libraries. We obtained a DME for each of these libraries, in the manner described previously.

Then, we obtained a prediction for the frequency of phenotypes in each of the three categories for the system mutant library consisting of low mutation probability *trans*- and high mutation probability *cis*-elements. To do so, we weighted the frequencies of phenotypes in each category of each partition combination library in *Figure 5* by the frequency of that partition in the original *trans*-element library (*Figure 2E*). For example, no expression *trans* +no expression *cis* library yielded 93.4% of phenotypes in 'no expression' category, and 6.6% of phenotypes in 'intermediate phenotype' category. These were weighted by 0.686, which is the frequency of phenotypes in no expression *trans*-element category from which this particular partition combination library was derived (*Figure 2E*). All weighted frequencies in the three categories – 'no expression phenotypes', 'intermediate phenotypes', and 'high expression phenotypes' – across all nine partition combination DMEs were added up to obtain a prediction for the distribution of phenotypes for the whole system library. As a control that it is the presence of the *cis* mutants that leads to a more accurate prediction of frequencies in the three categories, we used the frequencies of phenotypes based on sorting accuracy. These two predicted distributions (experimental prediction based on partition libraries and the prediction based on sorting accuracy) were compared to the actual distributions using a Pearson's Chi-squared test.

## Mathematically accounting for the genetic regulatory structure of the system

We tested if accounting for the genetic regulatory structure improved the naïve convolution-based prediction of the system DME. Similar to the experimental approach, we incorporated the knowledge of the effects of *cis* mutations in the absence of CI ($F^-_{cis}$). In addition to the previous analysis, we assume that *trans* mutants showing high expression phenotypes (namely, those mutants that have the same expression as the wildtype in the absence of CI) are loss-of-function mutants that do not bind any *cis* mutants. To incorporate this information into the convolution, we (i) removed the high expression peak from the *trans* DME; (ii) performed a convolution between the remainder of the *trans* DME and the $f_{cis}$, (iii) introduced a cutoff, and then, (iv) instead of adding back the high expression wildtype in the absence of CI ($F^-_{wt}$), we add the distribution of *cis* mutations in the absence of CI ($F^-_{cis}$). This distribution is, as for the naïve convolution, added in proportion to the removed high expression *trans* phenotypes to normalize the whole distribution. Then, we evaluated the difference between the predicted DME and the observed system DME using a Pearson's Chi-squared test, as previously described. We did this for the three system libraries shown in *Figure 2* and for the high mutation probability *cis* +low mutation probability *trans* library, shown in *Figure 4*.

## Not all intermolecular epistasis arises from the genetic regulatory structure of the system

While considering the effects of *cis* mutations on RNAP binding (and hence accounting for the genetic regulatory structure of the system) explained much of intermolecular epistasis we observed in system DMEs, we wanted to evaluate the extent to which other mechanisms might be contributing to epistasis between the *cis*- and the *trans*-element. To this end, we designed a library of 150 system double mutants, by combining point mutations in *cis*- and *trans*-elements with specific phenotypes. Namely, we selected 150 *trans* mutants that exhibited full repression, and 150 *cis*-element mutants that exhibited high expression in the absence of CI (*Figure 6—figure supplement 1*). The system double mutant library made in this manner corresponds to the partition combination shown in *Figure 5G*. Note that not all 150 double mutants had a unique point mutation in the *cis*-element, since we could not identify 150 mutations in *cis* that did not significantly affect expression levels in the absence of CI. Then, we measured fluorescence levels for all double mutants and their constitutive single mutants, and from those measurements calculated epistasis, in the same manner as described above. Finally, we tested if existence of epistasis depended on the location of point mutations (*Figure 6—figure supplement 2*; *Figure 6—source data 2*) with Pearson's Chi-squared test, as previously described.

## Acknowledgements

We thank N Barton, T Bergmiller, A Betancourt, K Bod'ova, C Igler, C Nizak, T Paixão, MPleska, D Siekhaus, M Steinrueck, and G Tkačik for their invaluable comments on the manuscript. This work was supported by the People Programme (Marie Curie Actions) of the European Union's Seventh Framework Programme (FP7/2007-2013) under REA grant agreement n° [291734] to ML and European Research Council under the European Union's H2020 Programme (FP/2007–2013)/ERC Consolidator Grant [n. 648440] to JPB.

## Additional information

### Funding

| Funder | Grant reference number | Author |
| --- | --- | --- |
| Seventh Framework Programme | 291734 | Mato Lagator |
| H2020 European Research Council | 648440 | Jonathan P Bollback |

The funders had no role in study design, data collection and interpretation, or the decision to submit the work for publication.

### Author contributions

Mato Lagator, Conceptualization, Data curation, Formal analysis, Funding acquisition, Validation, Investigation, Visualization, Methodology, Writing—original draft; Srdjan Sarikas, Formal analysis, Investigation, Writing—review and editing; Hande Acar, Conceptualization, Methodology, Writing—review and editing; Jonathan P Bollback, Conceptualization, Supervision, Funding acquisition, Project administration, Writing—review and editing; Călin C Guet, Conceptualization, Resources, Supervision, Funding acquisition, Project administration, Writing—review and editing

### Author ORCIDs

Mato Lagator [iD] http://orcid.org/0000-0001-7847-3594
Hande Acar [iD] http://orcid.org/0000-0003-1986-9753
Jonathan P Bollback [iD] https://orcid.org/0000-0002-4624-4612
Călin C Guet [iD] http://orcid.org/0000-0001-6220-2052

### Decision letter and Author response

Decision letter https://doi.org/10.7554/eLife.28921.036
Author response https://doi.org/10.7554/eLife.28921.037

## Additional files

### Supplementary files

• Transparent reporting form
DOI: https://doi.org/10.7554/eLife.28921.035

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
