## [Decision Letter]

Thank you for submitting your article "Intermolecular epistasis increases phenotypic variation in a gene regulatory system" for consideration by *eLife*. Your article has been reviewed by three peer reviewers and the evaluation has been overseen by Patricia Wittkopp as the Reviewing Editor and Senior Editor. The reviewers have opted to remain anonymous.

The reviewers have discussed the reviews with one another and the Reviewing Editor has assembled a list of concerns for you to consider.

Summary:

The creative approach to investigating an important question (interactions among genetic changes affecting *cis-* and *trans-*regulatory components of a regulatory network) was applauded, but concerns were raised by two of the reviewers about the null model used to infer the effects of epistasis. Following discussion, we all agreed this is a major concern since this test underlies the core observation and advance of the paper, and must be convincingly addressed. I should note that the remedy for this concern is not straight forward to us, and we anticipate that it will require a proof of principle experiment with a small number of mutants and/or a much more sophisticated mathematical/statistical treatment to better model the null hypothesis. We are concerned that sufficiently addressing this concern will take more than the 2 months typically allowed by *eLife* for revisions prior to publication and/or result in findings that are less novel, but have decided to extend an opportunity to try to address this concern in a response letter.

Instead of our usual policy of consolidating the remarks, I am attaching the full set of reviews because the concerns are explained in more detail there.

Essential revisions:

1) Revise the null model to better align with prior work and theory, possibly including some proof-of-principle tests of the strategy adopted.

2) Provide stronger support for the conclusion that the interactions are epistatic (non-additive).

Reviewer #1:

The manuscript uses random mutants in a *trans-*acting repressor and its *cis* target and finds epistasis such that the combined *cis* and *trans* effects are not predicted by their effects alone. The work is conceptually novel and the data provide new insight into how regulatory systems might evolve. My main concern is what the expected phenotype distribution should look like in the absence of epistasis. In part this could be due to a lack of clarity in the definition of epistasis or the clarity of writing.

1) Epistasis is defined differently in different fields. In quantitative genetics (as in this study) it is a non-additive interaction. It appears the assumption is that the *cis* and *trans* components are additive on a log scale in the absence of epistasis. However, for fitness the assumption is often that epistasis is a deviation from multiplicative effects. The definition should be clearly stated.

2) Certain epistatic relationships are entirely expected. For example, in the absence of CI the phenotype of the *cis* + *trans* mutants should equal that of *cis* mutants regardless of *trans* effect since *trans* mutants are not expressed. This is nicely observed in Figure 2—figure supplement 1.

3) The expected phenotype distribution in the absence of epistasis is not clear and perhaps incorrect. The interpretation of Figure 2 is that there are too many cells with intermediate expression phenotype than expected. However, under a simple additive case one should expect panel D (cis) + panel (G) *trans* to produce mostly intermediate phenotypes since the most common phenotype in panel D is low expression and the most common in panel G is high expression, and low + high = intermediate. That said, the authors recognize that high expression in panel G (trans) is likely non-functional CI. To account for this they subtract out the high expression of panel G using WT in the absence of CI. This doesn't make sense. When non-functional CI is combine with *cis* mutations one would expect to see a phenotype distribution of *cis* in the absence of CI, as shown in Figure 2—figure supplement 1, which is much more uniform and has a lot of intermediate frequency phenotypes. Eyeballing it looks like this would generate an expected pattern close to that observed a lot of intermediate phenotypes. The interpretation for the intermediate class is that mutant *Cis* are binding to mutant *cis-*elements. However, I don't think the authors have clearly shown that the increased frequency of the intermediate class is different from what one expect in the absence of epistasis.

There is not an obvious solution to calculating the expected phenotype distribution. The options I see are given below. However, engaging a mathematical biologist or statistician to appropriate generate expected phenotype distributions may yield better options.

a) average of *cis* and *trans.* Note that the methods stats that convolution of *cis* and *trans* yields phenotypes outside of biological range. The average of two numbers can't be above both numbers so the convolution must be *cis*+*trans* rather than (*cis*+*trans*)/2.

b) use a categorical model where *cis* + non-functional *trans* (high) = *cis* in the absence of CI, *cis*+ functional CI (low) = *cis*, *cis* + intermediate *trans* = weighted average of *cis* effects with CI and *cis* effects without CI with weighting depending on the whether intermediate is closer to low or high expression.

Figure 4 also shows expectations in the absence of epistasis that don't make sense. The results show that 7/9 system libraries show prevalent epistasis because adding in the *cis* mutants alters the phenotypic effects of the *trans* mutants. The figure and interpretation now seem to use a different definition of epistasis – the one used in classical genetics. In Figure 4 the grey shows unmodified *trans* and orange shows *trans* which is modified by *cis.* Panel G is an example where low *trans* is combined with high *cis* mutants. Unmodified (grey) is low, modified (orange) is intermediate or high. The observed distribution is spread across low, intermediate and high expression. But this is exactly what one would expect under an additive model of low + high effects.

4) The convolution of libraries to get an expectation. A gamma distribution was used for the *cis* library. It would be better to use the actual empirical distribution through random sampling of *cis* effects and *trans* effects with replacement. What assumption was made about the *trans* distribution? It is bimodal so not easily fit to a standard distribution. Is it a gamma after subtraction?

5) Noise in expression should be mentioned as it could contribute as well to the sorting accuracy statements.

6) Properties of mutant library. How was the average mutation frequency measured (1%-7%), and are mutants Poisson distributed. Simply using estimates provided by mutagenesis kit is not a sufficient measure of the library complexity. The manuscript states that 40 clones were Sanger sequenced. Is this 40 for each of the low, intermediate and high? What are the observed average number of mutants? Simply stating that they conform to the expected distribution given by the kit is not ok, you should use the empirical estimate obtained from sequencing.

7) What is the frequency of plasmids with no insert from cloning, either for the CI protein or the *cis* element? Typically this is low, but clones are confirmed this way. In high throughput experiments there will always be some frequency of plasmids ligated without an insert.

Reviewer #2:

Lagator et al. measure how mutations lead to phenotypic variation in gene expression at the systems levels. This minimal system based on phage lambda contains the CI repressor and a constitutive promoter driving *venus-yfp*. Three types of mutagenesis libraries were created: the '*cis*' library mutated the constitutive promoter, the 'trans' library mutated the protein coding sequence of the CI repressor, and the 'system' library mutated combined both sets of mutations. For each library flow cytometry was used to measure the Distribution of Mutational Effects (DME), the phenotypic variation in gene expression of the population. The main finding is that the quantitative shapes of the DME are different for the *cis*, *trans*, and combined libraries. In particular there is an excess of constructs in the combined library with intermediate expression levels. The authors claim epistasis between *cis* and *trans* mutations must be invoked to explain the intermediate level phenotypes in the DME of the combined libraries. The authors further discuss how phenotypically neutral mutations may express phenotypes in combination with other mutations.

I may be missing something here, but I think the main result of this manuscript may be trivial. There is a straw man hypothesis in the text which says that "the intuitive expectation (is) that an increase in the number of mutations ought to result in an increase in non-functional ('no expression') phenotypes". I agree that an increased mutation will lead to more loss of function mutations, but in this system loss of function *trans* mutants in CI increase expression while loss of function *cis* mutations in the promoter either decrease expression through decreased polymerase binding or increase expression through decreased CI binding. We might very well expect the combined library to have more intermediate phenotypes as loss of function mutations that both increase and decrease expression average each other out. One need not necessarily invoke epistasis to explain the increase in intermediate phenotypes in the combined library.

I also disagree with the primary interpretation that there are many "neutral" *cis* mutations that then manifest phenotypically in combination with a *trans* mutation. This is one plausible interpretation. An opposite interpretation is that there are many highly penetrant *trans* mutations (21.7% in Figure 2) and that in combination with a *cis* mutation the effects of these *trans* mutations are buffered. The 10% increase in intermediate phenotypes in Figure 2 almost exactly mirrors the 10% decrease in high expressing phenotypes. This suggests that a large fraction of intermediate phenotypes come from highly penetrant *trans* mutations being buffered by *cis* mutations, and not from silent *cis* mutations that interact with *trans* mutations. In other words the mass in the DME moves from the high expressing bin into the medium expressing bin, not from the low expressing bin into the medium bin.

Reviewer #3:

This manuscript describes the DME for interacting *cis-* and *trans-*regulatory sequences in a well-defined regulatory system. The primary finding is that epistatic interactions between mutations in these two components produce a larger range of phenotypes than variation in either single component. On the one hand, this type of epistasis is perhaps required to emerge from the known interactions of CI and the *cis-*sequence in the system. On the other hand, the quantitative consequences of this epistasis have rarely been described in detail and I think it is interesting to see how these interactions shape the phenotypic space explored. The use of mutant alleles with multiple mutations and the absence of any discussion of the identity of mutations mediating the observed epistasis that would have provided more insight into molecular mechanisms reduced my enthusiasm for this work, however. In addition, how much does intramolecular epistasis contribute to the patterns reported? One point where these questions are ameliorated is in the analysis of 109 single point mutations in *cis* and 73 in trans, but the locations of these changes with CI and the promoter are not described. Looking at the identity of these mutations in more detail might provide some insight into the specific interactions between *cis* and *trans* acting factors that produced the intermediate expression phenotypes.

[Editors' note: further revisions were requested prior to acceptance, as described below.]

Thank you for resubmitting your work entitled "Intermolecular epistasis increases phenotypic variation in a gene regulatory system" for further consideration at *eLife*. Your revised article has been favorably evaluated by three peer reviewers, and the evaluation was overseen byPatricia Wittkopp as the Reviewing and Senior Editor. The following individual involved in review of your submission has agreed to reveal his identity: Justin Fay.

The manuscript has been improved but there are some remaining issues that need to be addressed before acceptance, as outlined below:

We appreciate the authors response to the reviewer's comments and inclusion of additional data addressing the concerns raised. For example, the definitions of epistasis and the methodology used to compute the naive DME are now more clear and easier to understand. We also appreciated the comparison to an empirically derived DME that accounts for our molecular knowledge of the components of CI system in phage. We remain convinced this is an interesting dataset addressing an interesting question, but also remain concerned that the conclusions drawn depend on the assumptions of the model, some of which we think are not plausible. We also agree, however, that it is not clear what the "correct" set of assumptions should be, so are supportive of publication despite these concerns.

In light of this uncertainty, we think a modification of the title and adjustment of the conclusions is appropriate. For example, we think the title should convey that the structure of regulatory circuits determines patterns of epistasis rather than regulatory circuits generate lots of unexpected epistasis.

In addition, we ask that the authors clarify their work further (no new data is needed). For example, two areas that seem fundamental to understanding the paper are: Does low + low = high expression under a naive model? If so what does low + high equal under an additive model? I'm still not sure. Statements like: "increasing the number of mutated components should introduce additional constraints, limiting the variation accessible through mutation" remain confusing. I am including the full comments from reviewer #1 below because they explain these remaining questions more fully.

Reviewer #1:

In this resubmitted manuscript, the authors revised their analysis and included substantial additional data. Primarily, they measured expression from 150 point mutations along with their double mutants. Overall the manuscript is greatly improved: it is more clearly presented and the individual single double mutant assays provides much greater confidence in their main result – epistasis between *cis-trans* mutants such that mutant *trans-*elements can bind mutant *cis-*elements generating expression patterns not expected from either mutant alone. However, as brought up in the initial review, the calculation of the double mutant expectations is problematic. In part, this may be related to clarity/understanding, but it could also indicate a problem in how these expectations were calculated. The expectations that I find problematic occur in Figure 2, but also in the double mutants, Figure 3 along with Figure 2—figure supplement 2.

Overall, there are some strong indications of surprising epistasis. For me this came from looking at Figure 3—figure supplement 3 through Figure 3—figure supplement 5 showing both the doubles and singles. However, eyeballing it is not easy and it would be much easier to read using bargraphs of single, single, double (obs) and expected. The examples of doubles with negative epistasis (Figure 3—figure supplement 3) seem to be quite small deviations since they look simply like a combination of the two single mutants. However, the cases with positive epistasis are striking in that many show low + low = intermediate rather than low which is what I believe the expectation to be.

While the examples are nice, the main analysis contains expectations that I don't find logical for the naive analysis. "Increasing the number of mutated components should introduce additional constraints, limiting the variation accessible through mutation": I disagree, given two sources of variation, combining them will increase variation beyond each individual component.

Central to the calculation of epistasis is the use of the convolution of *cis* + *trans* effects to derive an expectation for the system (doubles). This expectation is shown in Figure 2. The question is what do we expect when we combine a low (*cis*) with either a low (trans) or high (trans) expression mutant. The convolution predicts this will mostly be high with a small amount of intermediate and low. Under a simple additive model one would expect low + low = low, and low + high = intermediate, which is quite similar to what is found. What is not clear to me is whether this is a problem in calculating the convolution of three DMEs or the assumptions in applying the convolution to get an expected level of expression. I think there must be clarity and agreement on what the expectation of low + high should be from the *cis* and *trans* library.

The 150 single mutants show similar patterns to what I would expect based on an additive model.

71/150 deviate from additive expectation. However, F2-2 shows that most single mutants have no effect (i.e. low expression). Why then do most of the observed doubles have an effect in the range of 1-3 when their effect should be zero? These observations are at odds with one another.

If the argument that high (trans) + low (cis) should be high expression because the repressor doesn't work, then this is exactly what one would expect if you include epistasis as a consequence of the way the regulatory system works and so is not really insightful. While this is a fine assumption to make later (non-naive), the simplest naive expectation needs to be understandable before making things more complicated.

Why didn't the positional information predict those that affect expression? One would expect that changes in binding sites for RNAP or CI would have quite different effects on expression.

---

## [Author Response]

Essential revisions:1) Revise the null model to better align with prior work and theory, possibly including some proof-of-principle tests of the strategy adopted.2) Provide stronger support for the conclusion that the interactions are epistatic (non-additive).

We would like to thank the Editors and the reviewers for dedicating their time and energy to our manuscript, and for providing us with insightful and detailed feedback and criticism that undoubtedly strengthened our work.

We previously provided a plan on how to tackle the problems identified by the reviewers. This resubmission contains all the experiments we promised to do, as well as some additional analyses that we did not mention in the plan and that we felt would make our claims substantially stronger (for example, the analyses of the relationship between epistasis and location of mutations, and the convolutions that account for the genetic structure of the system). Together, the new evidence provides strong support for the existence of intermolecular epistasis in our system and demonstrates that the effects of such epistasis on shaping the evolution of transcriptional regulatory systems are significant.

In the revised version of the manuscript, we addressed the two essential revisions in the following ways:

1) We address the first essential revision by explaining in more detail why convolution is the correct null model against which to estimate epistasis for a DME. In particular, we outline how a prediction of epistasis in the absence of any knowledge of the underlying molecular mechanisms (which is what a ‘naïve’ convolution in the manuscript calculates) is meaningful. We were encouraged by our discussions with several mathematical biologists and population geneticists, including Nick H. Barton, who confirmed that convolution is the standard approach to estimating combined DMEs.

2) While the deviation of observed DMEs from those predicted by convolutions provides evidence for the existence of epistasis between *cis-* and *trans-*elements, we included additional experiments to better assess how prevalent such interactions are, and to determine their molecular and mechanistic origins. Specifically, we addressed the second essential revision more thoroughly by creating two additional libraries, each consisting of 150 double mutants and their corresponding single mutants. These libraries allowed us to estimate how common intermolecular epistasis is, and how frequently do epistatic interactions arise that cannot be explained by the underlying genetic regulatory structure. As we now know the identity of these mutants, this allowed us to analyze the relationship between the location of mutations and epistasis.

3) By conducting convolution predictions of the system DMEs that account for the underlying genetic regulatory structure, which allowed us to demonstrate that much of intermolecular epistasis comes from the mechanism of how the system functions. This analysis was done as a complement to the experimental approach of combining library partitions (old Figure 4, which is now Figure 5).

Reviewer #1:The manuscript uses random mutants in a trans-acting repressor and its *cis* target and finds epistasis such that the combined *cis* and *trans* effects are not predicted by their effects alone. The work is conceptually novel and the data provide new insight into how regulatory systems might evolve. My main concern is what the expected phenotype distribution should look like in the absence of epistasis. In part this could be due to a lack of clarity in the definition of epistasis or the clarity of writing.1) Epistasis is defined differently in different fields. In quantitative genetics (as in this study) it is a non-additive interaction. It appears the assumption is that the *cis* and *trans* components are additive on a log scale in the absence of epistasis. However, for fitness the assumption is often that epistasis is a deviation from multiplicative effects. The definition should be clearly stated.

The reviewer is right to point out the lack of clarity in our definition of epistasis. Throughout the manuscript, our phenotype of interest is log_10_ of fluorescence, and we calculate epistasis as the deviation from the additive assumption. This means that, on a linear scale, we calculate epistasis as a deviation from the multiplicative assumption. In the two libraries of 150 double mutants, we calculate epistasis as the deviation of the log_10_ of wildtype-normalized fluorescence of the double mutant from the sum of the log_10_ of wildtype-normalized fluorescence of the two single mutants. To predict the DME of the system through a convolution, we adopt the same approach. As such, we employ the same null model for both types of data. Throughout the manuscript we improved the clarity of how we define epistasis (in particular, see section ‘intermolecular epistasis drives the increase in phenotypic variation’, and the Materials and methods sections ‘Naïve convolution of component distributions as the null model for additivity between mutations’, and ‘Epistasis between random point mutants’).

2) Certain epistatic relationships are entirely expected. For example, in the absence of CI the phenotype of the *cis* + *trans* mutants should equal that of *cis* mutants regardless of trans effect since trans mutants are not expressed. This is nicely observed in Figure 2—figure supplement 1.

Figure 2—figure supplement 1 shows DMEs in the absence of CI. Of course, as the reviewer pointed out, in such conditions, the three *trans* mutation libraries (panels E,F,G) should be indistinguishable from the wildtype in panel A, and we show them primarily as a consistency check that none of the *trans* mutants have any effect in this environment. In the language of convolutions, the underlying “true” distribution of our *trans* library here is a δ-function, a unit element for the operation of convolution. Convolving any distribution with it yields the same distribution. This interpretation is consistent with the observation that *cis* DMEs in Figure 2—figure supplement 1 are virtually indistinguishable from system DMEs. We discuss these observations in subsection “Naïve convolution of component distributions as the null model for additivity between mutations”.

3) The expected phenotype distribution in the absence of epistasis is not clear and perhaps incorrect. The interpretation of Figure 2 is that there are too many cells with intermediate expression phenotype than expected. However, under a simple additive case one should expect panel D (*cis*) + panel (G) trans to produce mostly intermediate phenotypes since the most common phenotype in panel D is low expression and the most common in panel G is high expression, and low + high = intermediate. That said, the authors recognize that high expression in panel G (*trans*) is likely non-functional CI. To account for this they subtract out the high expression of panel G using WT in the absence of CI. This doesn't make sense. When non-functional CI is combine with *cis* mutations one would expect to see a phenotype distribution of *cis* in the absence of CI, as shown in Figure 2—figure supplement 1, which is much more uniform and has a lot of intermediate frequency phenotypes. Eyeballing it looks like this would generate an expected pattern close to that observed a lot of intermediate phenotypes. The interpretation for the intermediate class is that mutant Cis are binding to mutant cis-elements. However, I don't think the authors have clearly shown that the increased frequency of the intermediate class is different from what one expect in the absence of epistasis.

In the new version of the manuscript, we distinguish between two approaches to performing a convolution. One is done in the absence of any knowledge of the genetic regulatory structure of the system. This approach is equivalent, conceptually, to how epistasis would be calculated for mutations in, for example, a protein, where there is little underlying mechanistic understanding of what those mutations might do. Note that it is deviations from this kind of a model, rather than a more complex one that accounts for the regulatory structure, that have a well-documented impact on organismal evolution. For this reason, we rely on this ‘naïve’ convolution as the null model against which to estimate epistasis (subsection “Intermolecular epistasis arises from the genetic regulatory structure of the system” and “Naïve convolution of component distributions as the null model for additivity between mutations”).

The second approach relies on the reviewer’s suggestion to “put back” the population of non-functional CIs, after convolution, as a distribution of *cis* in the absence of CI, which is shown in Figure 2—figure supplement 1. We use this approach to demonstrate how a good portion of intermolecular epistasis can be attributed to the underlying molecular mechanism of how the system functions (by which we simply mean that a high expression *trans* mutation is a loss of function mutation, whose phenotype in the system would be that of the *cis* mutant in the absence of CI). Indeed, as the reviewer hypothesized, doing the convolutions in this manner results in a more accurate prediction of the system distribution (subsection “Intermolecular epistasis arises from the genetic regulatory structure of the system” and “Mathematically accounting for the genetic regulatory structure of the system”). We would like to thank the reviewer for this very helpful suggestion.

The justification for why we removed (and then put back) the high expression peak of the *trans* library even for the ‘naïve’ convolution is to prevent the predicted DME from having expression levels outside the biologically realistic range. In particular, the upper limit to expression is set by the strength of the P_R_ promoter, which is one of the strongest known promoters in *E. coli*. Therefore, by removing the high expression peak before convolution we simply impose this limit on the predicted system DME. This approach is biologically motivated, as high expression *trans* mutants are loss of function mutants (subsection “Naïve convolution of component distributions as the null model for additivity between mutations”).

There is not an obvious solution to calculating the expected phenotype distribution. The options I see are given below. However, engaging a mathematical biologist or statistician to appropriate generate expected phenotype distributions may yield better options.a) average of *cis* and *trans*. Note that the methods stats that convolution of *cis* and *trans* yields phenotypes outside of biological range. The average of two numbers can't be above both numbers so the convolution must be cis+trans rather than (cis+trans)/2.b) use a categorical model where *cis* + non-functional *trans* (high) = cis in the absence of CI, cis+ functional CI (low) = *cis*, *cis* + intermediate *trans* = weighted average of *cis* effects with CI and *cis* effects without CI with weighting depending on the whether intermediate is closer to low or high expression.

In discussions with several mathematical biologists and population geneticists, including Nick Barton, we have come to be persuaded that the right null model is the ‘naïve’ convolution, as it is the mathematical equivalent of the standard definition of epistasis as ε = m_12_ / (m_1_ x m_2_). We hope that the explanations we have provided in the manuscript (subsection “Intermolecular epistasis arises from the genetic regulatory structure of the system” and “Naïve convolution of component distributions as the null model for additivity between mutations”) and in our answer to the previous part of reviewer 1 – comment 3 justify our choice and persuade the reviewer that the convolution is, indeed, the correct null model against which to evaluate the presence of epistasis.

Figure 4 also shows expectations in the absence of epistasis that don't make sense. The results show that 7/9 system libraries show prevalent epistasis because adding in the *cis* mutants alters the phenotypic effects of the trans mutants. The figure and interpretation now seem to use a different definition of epistasis – the one used in classical genetics. In Figure 4 the grey shows unmodified trans and orange shows trans which is modified by *cis*. Panel G is an example where low trans is combined with high *cis* mutants. Unmodified (grey) is low, modified (orange) is intermediate or high. The observed distribution is spread across low, intermediate and high expression. But this is exactly what one would expect under an additive model of low + high effects.

We fully agree with the reviewer’s comments – we did switch our definition of epistasis at this point (unfortunately, without realizing it ourselves!), and had done so in a way that was not consistent. Specifically, calculating epistasis as the deviation from a multiplicative expectation of two DMEs obtained in different environments that can never coexist does not make sense. Thanks to the reviewer’s comment, we have excluded such analysis from the data presented in Figure 4 (now Figure 5). Instead, we utilized this data only to demonstrate how linking the genetic structure of a regulatory system can help elucidate the molecular basis of epistasis. Because by “genetic structure of a system” we specifically mean accounting for the effects of mutations in *cis* in the absence of CI, we combined the *cis* library in the absence of CI with the *trans* library in the presence of CI.

4) The convolution of libraries to get an expectation. A gamma distribution was used for the *cis* library. It would be better to use the actual empirical distribution through random sampling of *cis* effects and trans effects with replacement. What assumption was made about the trans distribution? It is bimodal so not easily fit to a standard distribution. Is it a gamma after subtraction?

Convolving empirical distributions directly suffers from two problems: 1) we would need to deconvolve noise from one of them; 2) convolution is highly unstable numerically and yields non-interpretable results. Having at least one distribution in its analytic form remedies numerical problems. In order to obtain an analytical form of one distribution (*cis* in our case), we “reverse engineered” it, so that the convolution of the analytical form with the wildtype distribution matches the observed *cis* distribution as closely as possible. We changed the text in order to better explain this point (subsection “Naïve convolution of component distributions as the null model for additivity between mutations”).

5) Noise in expression should be mentioned as it could contribute as well to the sorting accuracy statements.

We agree with the reviewer, and have now conducted additional experiments to understand how gene expression noise might be affecting our findings. By observing gene expression levels in 180 monoclonal isolates from experimental mutant libraries and in 180 single/double mutants, we tested if gene expression noise depended on: (i) gene expression levels; (ii) frequency of mutations (single, double, low, intermediate, and high mutation frequency); and (iii) location of mutations (*cis, trans*, or system). We found no differences in gene expression noise between isolated mutants. We describe these findings in the manuscript (subsection “Mutating the whole system increases phenotypic variation” and “Estimating gene expression noise from flow cytometry measurements”).

6) Properties of mutant library. How was the average mutation frequency measured (1%-7%), and are mutants Poisson distributed. Simply using estimates provided by mutagenesis kit is not a sufficient measure of the library complexity. The manuscript states that 40 clones were Sanger sequenced. Is this 40 for each of the low, intermediate and high? What are the observed average number of mutants? Simply stating that they conform to the expected distribution given by the kit is not ok, you should use the empirical estimate obtained from sequencing.

We apologise for the lack of clarity on this issue. We sequenced 40 clones from each *cis* and each *trans* library, and found that the distribution of mutations for each library is not significantly different from the expected distribution. More details can be found in subsection “Creating the mutant libraries”, as well as in Figure 2—source data 2.

7) What is the frequency of plasmids with no insert from cloning, either for the CI protein or the *cis* element? Typically this is low, but clones are confirmed this way. In high throughput experiments there will always be some frequency of plasmids ligated without an insert.

We understood that there are two concerns expressed by the reviewer.

(i) *what is the proportion of re-ligated wildtype plasmids, in which the mutated region was not inserted and instead the wildtype used as the cloning template re-ligated back*: Following from comment 6) above, by observing the frequency of wildtype genotypes in the sequenced *trans* libraries (which, due to the relatively large number of mutations should not contain any wildtype sequences), we estimate that <5% of each library is re-ligated with the wildtype insert.

(ii) *what is the proportion of cloned plasmids without any insert at all*: In 6x40 sequenced clones, we did not identify any in which there was no insert at all – either the wildtype was or was not replaced by a mutant genotype. We updated the manuscript to be clear about this. This is most likely because we used two pairs of non-complimentary restriction sites for cloning the *cis-* and the *trans-*element mutants. We added this information in the text (“subsection “Creating the mutant libraries”).

Reviewer #2:[…] I may be missing something here, but I think the main result of this manuscript may be trivial. There is a straw man hypothesis in the text which says that "the intuitive expectation (is) that an increase in the number of mutations ought to result in an increase in non-functional ('no expression') phenotypes.". I agree that an increased mutation will lead to more loss of function mutations, but in this system loss of function trans mutants in CI increase expression while loss of function *cis* mutations in the promoter either decrease expression through decreased polymerase binding or increase expression through decreased CI binding. We might very well expect the combined library to have more intermediate phenotypes as loss of function mutations that both increase and decrease expression average each other out. One need not necessarily invoke epistasis to explain the increase in intermediate phenotypes in the combined library.

We agree with the reviewer that we need to better demonstrate that epistasis is indeed present in our system. We think that the additional analyses of DMEs through the two types of convolutions (the naïve convolution, which is the right null model for no epistasis, is shown in Figure 2; while the convolutions that account for the genetic structure of the system are shown in Figure 2—figure supplement 6), as well as the two new libraries of single/double mutants (Figure 3 and Figure 6), persuade the reviewer of the ubiquitous presence of intermolecular epistasis in our system. In the new version of the manuscript, we demonstrate that much of intermolecular epistasis comes from an obvious interaction – what we term ‘genetic regulatory structure of the system’. We demonstrate this experimentally and mathematically (subsection “Intermolecular epistasis arises from the genetic regulatory structure of the system”). We also include a discussion on the fact that deviations from a naïve additive assumption for interactions between mutations are what shape the adaptive landscape. Throughout the manuscript, we now include a more in depth discussion of all of the points raised in this comment (especially in the section ‘intermolecular epistasis drives the increase in phenotypic variation’,). We also removed the identified line (‘the intuitive expectation’).

I also disagree with the primary interpretation that there are many "neutral" *cis* mutations that then manifest phenotypically in combination with a trans mutation. This is one plausible interpretation. An opposite interpretation is that there are many highly penetrant trans mutations (21.7% in Figure 2) and that in combination with a *cis* mutation the effects of these trans mutations are buffered. The 10% increase in intermediate phenotypes in Figure 2 almost exactly mirrors the 10% decrease in high expressing phenotypes. This suggests that a large fraction of intermediate phenotypes come from highly penetrant trans mutations being buffered by *cis* mutations, and not from silent *cis* mutations that interact with trans mutations. In other words the mass in the DME moves from the high expressing bin into the medium expressing bin, not from the low expressing bin into the medium bin.

The reviewer is right to point out that at least a part of the results we observe is due to penetrant *trans* mutations, which lead to loss of function (and hence are not neutral), and that consequently change phenotype based only on the *cis* background. However, by using the library of 150 double mutants, we evaluated the relative importance of these two forces, and find that neutral *trans* mutations are more common than penetrant ones (Figure 3). In fact, around 75% of all double mutants that are in significant epistasis have a neutral, or near-neutral *trans* mutation (subsection “Intermolecular epistasis drives the increase in phenotypic variation2”). We also include a discussion on the limitations of using single and double mutants to assign specific importance to one of these mechanisms over the other (neutral vs penetrant *trans* mutants) when considering mutants with a greater number of mutations, such as for the distributions of mutant libraries shown in Figure 2 (Discussion paragraph three).

Reviewer #3:This manuscript describes the DME for interacting *cis*- and *trans*-regulatory sequences in a well-defined regulatory system. The primary finding is that epistatic interactions between mutations in these two components produce a larger range of phenotypes than variation in either single component. On the one hand, this type of epistasis is perhaps required to emerge from the known interactions of CI and the *cis*-sequence in the system. On the other hand, the quantitative consequences of this epistasis have rarely been described in detail and I think it is interesting to see how these interactions shape the phenotypic space explored. The use of mutant alleles with multiple mutations and the absence of any discussion of the identity of mutations mediating the observed epistasis that would have provided more insight into molecular mechanisms reduced my enthusiasm for this work, however.

While our focus in the original submission was to describe the phenotypic consequences of intermolecular epistasis, we agree with the reviewer that our findings would be strengthened if we could provide some detail on the kinds of mutations that result in the observed patterns. In the mutant libraries used in the original submission, the mutation numbers were too high to do this. Therefore, we created two new mutant libraries with 150 double mutants and their corresponding single mutants. For every single and double mutant in these libraries we know the identity and location of mutations (Figure 3—figure supplement 1; Figure 3—source data 1; Figure 6—figure supplement 2; Figure 6—source data 2). We analyze this data to identify if the location of mutations has any effect on the existence of epistasis, and find that much of negative epistasis in the system stems from loss-of-function *trans* mutations. Maybe more interestingly, the existence of interactions that *cannot* be explained by the underlying regulatory structure of the system are more likely to be found in the linker region of the repressor CI (“Accounting for the genetic regulatory structure of the system does not explain all intermolecular epistasis”).

In addition, how much does intramolecular epistasis contribute to the patterns reported?

The reviewer is right to ask about intramolecular epistasis. Of course, there must be interactions between mutations in *cis* alone, as well as between mutations in *trans* alone. For our findings to be valid, we assume that the nature of interactions between mutations in *cis* does not depend on the *trans* background. Conversely, we also assume that the epistasis between *trans* mutations is independent of mutations in *cis*. In other words, we assume that intramolecular epistasis in each component is independent of the mutations in the other component. We cannot conceive of an experimental design by which to test these assumptions, and as such are limited to only discussing them. Having said that, we hope that some of these concerns are addressed by looking at the two 150 system single/double mutant libraries, since in those double mutants there is no intramolecular epistasis (as one point mutation is in *cis* and the other in *trans*).

One point where these questions are ameliorated is in the analysis of 109 single point mutations in *cis* and 73 in *trans*, but the locations of these changes with CI and the promoter are not described. Looking at the identity of these mutations in more detail might provide some insight into the specific interactions between *cis* and *trans* acting factors that produced the intermediate expression phenotypes.

With the expansion of the single mutant libraries, and the addition of two system double mutant libraries consisting of 150 unique double mutants, we provide insight into where the mutations that lead to an interaction might be located in the *cis* and the *trans-*element. For 150 random double mutants, we do not find any relationship between epistasis and the location of mutations (subsection “Intermolecular epistasis drives the increase in phenotypic variation”). However, we find that the positive epistasis that emerges from a novel binding property of a repressor mutant is associated with the linker region connecting the DNA binding domain and the dimerization domain of the protein (subsection “Accounting for the genetic regulatory structure of the system does not explain all intermolecular epistasis”).

[Editors' note: further revisions were requested prior to acceptance, as described below.]

We appreciate the authors response to the reviewer's comments and inclusion of additional data addressing the concerns raised. For example, the definitions of epistasis and the methodology used to compute the naive DME are now more clear and easier to understand. We also appreciated the comparison to an empirically derived DME that accounts for our molecular knowledge of the components of CI system in phage. We remain convinced this is an interesting dataset addressing an interesting question, but also remain concerned that the conclusions drawn depend on the assumptions of the model, some of which we think are not plausible. We also agree, however, that it is not clear what the "correct" set of assumptions should be, so are supportive of publication despite these concerns.In light of this uncertainty, we think a modification of the title and adjustment of the conclusions is appropriate. For example, we think the title should convey that the structure of regulatory circuits determines patterns of epistasis rather than regulatory circuits generate lots of unexpected epistasis.

We thank the editors and the reviewers for suggesting a modification to the title, which we think now better captures the main point of our work. As the reviewers and editors pointed out, whether an increase in intermediate phenotypes can be attributed to epistasis is after all dependent on how one calculates/defines epistasis. What our work shows is that accounting for the underlying molecular mechanisms (in our case in the form of the structure of the gene regulatory network) can help to better define how mutations interact with each other. The new title should emphasize this point more clearly. In addition, we modified the Discussion section to emphasize how our results indicate that null models of epistasis can be inaccurate.

In addition, we ask that the authors clarify their work further (no new data is needed). For example, two areas that seem fundamental to understanding the paper are: Does low + low = high expression under a naive model? If so what does low + high equal under an additive model? I'm still not sure.

We made several changes to the manuscript in order to better clarify the naïve null model used in our study. Most importantly, we introduced a new table (Table 1), which shows the predictions not only for ‘low’ – ‘low’ and ‘low’ – ‘high’, but for how all combinations of the three categorical mutational effects (those leading to ‘no’, ‘intermediate’, or ‘high’ expression phenotypes) interact under an additive null model. This table provides an intuitive understanding for how the additive null model works when considering distributions of mutational effects.

Furthermore, we recognize from the comments of reviewer #1 that we were not clear in our description of how we convolved two distributions. We convolved the observed *trans* DME (as shown in Figure 2) with the ‘true’ distribution of effects in *cis*. This ‘true’ distribution shows how mutations in *cis* modify wildtype expression levels – and effectively all mutations in *cis* are either neutral or increase expression (as seen in Figure 2—figure supplement 7). Because of this, convolving a low expression *trans* distribution with the ‘true’ *cis* distribution would result in an increase in intermediate and high expression phenotypes, because mutations in *cis* are either neutral or increase expression. Similarly, convolving a high expression *trans* DME with the ‘true’ *cis* distribution would result in only high expression phenotypes. We have introduced changes both in the main text (subsection “Intermolecular epistasis drives the increase in phenotypic variation”) and in the Materials and methods, in order to further clarify how the predictions by the ‘naïve’ convolution were made.

Statements like: "increasing the number of mutated components should introduce additional constraints, limiting the variation accessible through mutation" remain confusing. I am including the full comments from reviewer #1 below because they explain these remaining questions more fully.

We did not intend this line to describe anything about the null model we used, but rather only as an illustration of how surprising our results were to us. We now understand how it could have caused confusion, and have as such removed it entirely.

Reviewer #1:In this resubmitted manuscript, the authors revised their analysis and included substantial additional data. Primarily, they measured expression from 150 point mutations along with their double mutants. Overall the manuscript is greatly improved: it is more clearly presented and the individual single double mutant assays provides much greater confidence in their main result – epistasis between *cis*-*trans* mutants such that mutant trans-elements can bind mutant *cis*-elements generating expression patterns not expected from either mutant alone.

We are glad to see that our new experimental data on the libraries of 150 single/double point mutations has greatly improved our main results and we thank the reviewer for the encouraging feedback.

However, as brought up in the initial review, the calculation of the double mutant expectations is problematic. In part, this may be related to clarity/understanding, but it could also indicate a problem in how these expectations were calculated. The expectations that I find problematic occur in Figure 2, but also in the double mutants, Figure 3 along with Figure 2—figure supplement 2.

Throughout the manuscript we define epistasis as the deviation from the additive assumption (given that we use log_10_ of expression as our phenotype of interest). Given that the wildtype used in this study has no measurable expression, effectively all mutations (in either *cis* or *trans*) are either neutral or increase expression. According to the additive null model, if an effect of a mutation is positive, in the absence of epistasis that effect should remain positive and have the same magnitude irrespective of the background. We have introduced Table 1 to provide an illustration of this null model by showing its predictions on three categorical phenotypes (no, intermediate, and high expression). We made substantial changes to the text to improve clarity of how we calculate the null model for DMEs shown in Figure 2 (subsection “Intermolecular epistasis drives the increase in phenotypic variation”; “Naïve convolution of component distributions as the null model for additivity between mutations”). We also clarified that, when calculating epistasis of double mutants shown in Figure 3, we use wildtype-normalized single mutant effects (subsection “Intermolecular epistasis drives the increase in phenotypic variation”).

Overall, there are some strong indications of surprising epistasis. For me this came from looking at Figure 3—figure supplement 3 through Figure 3—figure supplement 5 showing both the doubles and singles. However, eyeballing it is not easy and it would be much easier to read using bargraphs of single, single, double (obs) and expected.

The primary purpose of including Figure 3—figure supplement 3 through Figure 3—figure supplement 5 was to demonstrate that estimating epistasis from flow cytometry measurements was the same as from plate reader measurements (as shown in Figure 3—figure supplement 6, which was previously Figure 3—figure supplement 2), therefore justifying our use of plate reader measurements for the two single/double mutant libraries. Because the two measurements are equivalent, we now include bar charts for all 150 random double mutants and their corresponding single mutants, measured in the plate reader (Figure 3—figure supplement 2).

The examples of doubles with negative epistasis (Figure 3—figure supplement 3) seem to be quite small deviations since they look simply like a combination of the two single mutants. However, the cases with positive epistasis are striking in that many show low + low = intermediate rather than low which is what I believe the expectation to be.While the examples are nice, the main analysis contains expectations that I don't find logical for the naive analysis. "Increasing the number of mutated components should introduce additional constraints, limiting the variation accessible through mutation": I disagree, given two sources of variation, combining them will increase variation beyond each individual component.

We agree with the reviewer that the instances of positive epistasis are usually of higher magnitude, and as such are more striking.

We apologize for the lack of clarity for the purpose of this sentence in the manuscript. It does not describe our null model in any way. It was simply there to illustrate why we originally found the results surprising, to contrast our results to some intuitive expectation that we had prior to actually conducting experiments. Given that this sentence served such a minor purpose, and that it resulted in a more serious confusion, we removed it from the manuscript.

Central to the calculation of epistasis is the use of the convolution of *cis* + *trans* effects to derive an expectation for the system (doubles). This expectation is shown in Figure 2. The question is what do we expect when we combine a low (*cis*) with either a low (trans) or high (trans) expression mutant. The convolution predicts this will mostly be high with a small amount of intermediate and low. Under a simple additive model one would expect low + low = low, and low + high = intermediate, which is quite similar to what is found. What is not clear to me is whether this is a problem in calculating the convolution of three DMEs or the assumptions in applying the convolution to get an expected level of expression. I think there must be clarity and agreement on what the expectation of low + high should be from the *cis* and *trans* library.

We now realize that the revised version of the manuscript was lacking clarity in our description of how we carried out the convolution. We do not convolve two distributions as shown in Figure 2. Instead, we convolve the *trans* distribution shown in Figure 2 with the ‘true’ *cis* distribution. This ‘true’ distribution is nothing more than a *cis* distribution (shown in Figure 2) relative to the wildtype, which shows how mutations in *cis* alter wildtype expression. As shown in Figure 2—figure supplement 7, mutations in *cis* either do not affect wildtype expression (are effectively neutral), or increase it. Because of this, convolving the ‘true’ *cis* distribution with the hypothetical low *trans* distribution would result in an increase in the frequency of intermediate and high expression mutants. This is one of the reasons why convolution predictions seen in Figure 2 contain a relatively high frequency of high expression mutants. The second reason is that we introduce a threshold to high expression, by setting the maximum possible expression to that of the wildtype in the absence of CI. The fact that the *P_R_* promoter is one of the strongest known promoters justifies this threshold. Without this threshold, convolving high expression *trans* mutants with the ‘true’ *cis* distribution would result in many mutants with higher expression than the wildtype. With the threshold (as explained in the Materials and methods section “Naïve convolution of component distributions as the null model for additivity between mutations”), convolution of high expression *trans* mutants with the ‘true’ *cis* distribution results in only high expression phenotypes.

For these two reasons, convolving the *trans* DME with a ‘true’ *cis* distribution, which contains only neutral mutations and mutations that increase expression, results in a prediction that has an increase in the frequency of intermediate and high expression mutants, while showing a decrease in the frequency of low expression mutants. We now include a new table (Table 1), which summarizes how the three categories of mutations (no, intermediate, and high expression mutations) interact under the additive null model. We also provide better explanations of how we utilized convolutions in the main text (subsection “Intermolecular epistasis drives the increase in phenotypic variation”) and in the Materials and methods.

The 150 single mutants show similar patterns to what I would expect based on an additive model. 71/150 deviate from additive expectation. However, F2-2 shows that most single mutants have no effect (i.e. low expression). Why then do most of the observed doubles have an effect in the range of 1-3 when their effect should be zero? These observations are at odds with one another.

We apologize for the confusion and the lack of detail in the way we presented the single/double mutant data. We introduce this dataset of random 150 double mutants in order to demonstrate that the existence of intermolecular epistasis, identified through a deviation from the convolution prediction, is also present in our system when only a single point mutation is introduced in each of the components. We find that combinations of point mutations in *cis* and *trans*, which mainly result in low expression phenotypes, sometimes have higher expression when combined. This increase is due to intermolecular epistasis between them, which we calculate as the deviation of the observed double mutant effect (normalized by the wildtype) from the predicted effect (wildtype-normalized *cis* mutant * wildtype-normalized *trans* mutant effect). In order to further clarify and provide information on how single mutations interact, we have now added a new figure (Figure 3—figure supplement 2), which contains the data for each double mutant, its corresponding single mutants, and the prediction based on the additive model.

On a mechanistic level, one can hypothesize how these interactions arise. Significant epistasis can be observed when a high expression *trans* mutant results in a low or intermediate expression system phenotype, when combined with a *cis* mutation.This kind of epistasis contributes to the observed increase in intermediate phenotypes in the system. In addition, some double mutants are in positive epistasis because *trans* mutants can fully bind the wildtype *cis*, while binding the mutated *cis* less well than the wildtype *trans* does. This scenario would also result in intermediate phenotypes arising from a combination of low *cis* + low *trans* mutations. The introduction of the new figure showing single mutant effects, as well as the predicted and the observed double mutant effects for all 150 random mutants (Figure 3—figure supplement 2) provides further insight into how each pair of mutations interacts specifically.

If the argument that high (*trans*) + low (*cis*) should be high expression because the repressor doesn't work, then this is exactly what one would expect if you include epistasis as a consequence of the way the regulatory system works and so is not really insightful. While this is a fine assumption to make later (non-naive), the simplest naive expectation needs to be understandable before making things more complicated.

In obtaining the ‘naïve’ convolution, we do not make any assumptions about the underlying nature of high expression *trans* mutants. We removed the sentence in the Materials and methods section“Naïve convolution of component distributions as the null model for additivity between mutations” that could have been misinterpreted as suggesting that we make such an assumption. We have previously included this sentence only to provide another biological justification for the threshold we impose on high expression phenotypes. Our argument for why high *trans* convolved with the ‘true’ *cis* distribution results in high expression phenotypes is because mutations in *cis* are either neutral or increase the expression level. We now clarify this point in the main text (subsection “Intermolecular epistasis drives the increase in phenotypic variation”), in the Materials and methods, and in Table 1.

Why didn't the positional information predict those that affect expression? One would expect that changes in binding sites for RNAP or CI would have quite different effects on expression.

We only studied whether the location of mutations had an effect on the existence and magnitude of epistasis, and not on whether different positions in the binding site have different effects on expression. We now clarify this point in the resubmission (subsection “Intermolecular epistasis drives the increase in phenotypic variation”).